# Molecular Regulation of Palatogenesis and Clefting: An Integrative Analysis of Genetic, Epigenetic Networks, and Environmental Interactions

**DOI:** 10.3390/ijms26031382

**Published:** 2025-02-06

**Authors:** Hyuna Im, Yujeong Song, Jae Kyeom Kim, Dae-Kyoon Park, Duk-Soo Kim, Hankyu Kim, Jeong-Oh Shin

**Affiliations:** 1Department of Anatomy, College of Medicine, Soonchunhyang University, Cheonan 33151, Republic of Koreamdeornfl@sch.ac.kr (D.-K.P.); dskim@sch.ac.kr (D.-S.K.); 2Department of Food and Biotechnology, Korea University, Sejong 339770, Republic of Korea; 3Department of Health Behavior and Nutrition Sciences, University of Delaware, Newark, DE 19711, USA

**Keywords:** palate development, congenital disorder, cleft palate/lip, genetic network, epigenetics, environmental factors

## Abstract

Palatogenesis is a complex developmental process requiring temporospatially coordinated cellular and molecular events. The following review focuses on genetic, epigenetic, and environmental aspects directing palatal formation and their implication in orofacial clefting genesis. Essential for palatal shelf development and elevation (TGF-β, BMP, FGF, and WNT), the subsequent processes of fusion (SHH) and proliferation, migration, differentiation, and apoptosis of neural crest-derived cells are controlled through signaling pathways. Interruptions to these processes may result in the birth defect cleft lip and/or palate (CL/P), which happens in approximately 1 in every 700 live births worldwide. Recent progress has emphasized epigenetic regulations via the class of non-coding RNAs with microRNAs based on critically important biological processes, such as proliferation, apoptosis, and epithelial–mesenchymal transition. These environmental risks (maternal smoking, alcohol, retinoic acid, and folate deficiency) interact with genetic and epigenetic factors during palatogenesis, while teratogens like dexamethasone and TCDD inhibit palatal fusion. In orofacial cleft, genetic, epigenetic, and environmental impact on the complex epidemiology. This is an extensive review, offering current perspectives on gene-environment interactions, as well as non-coding RNAs, in palatogenesis and emphasizing open questions regarding these interactions in palatal development.

## 1. Introduction

Craniofacial development, especially palatogenesis, is among the intricate processes in vertebrate embryogenesis, and requires the precise coordination of numerous cellular and molecular events. Secondary palate formation is critical, because its disruption results in cleft lip and/or palate (CL/P), a prevalent congenital anomaly in approximately 1/700 live births across populations. Development of the palate is a complicated process that requires neural crest derived cells to coordinate cellular processes such as growth, movement, differentiation, and cell death. Palatogenesis is controlled by conserved signaling pathways such as TGF-β, BMP, FGF, WNT, and SHH. These pathways help the palatal shelf grow, rise, and fuse, similar to those of other organ systems. These processes function as part of a network that controls genes and epigenetics. Epigenetic processes, including DNA methylation, histone changes, and non-coding RNAs, have been linked to essential aspects of palatal development owing to progress in molecular biology. When a woman is early in her pregnancy and is exposed to factors such as smoking, drinking, retinoic acid (RA), and some teratogens, they can change the genetic and epigenetic processes that affect palatal development. It is essential to know their complex interactions to design more efficient preventive and therapeutic strategies for orofacial clefts (OFCs). This review provides a thorough overview of the biology of secondary palate development, emphasizing the synergy and interaction between genetic, epigenetic, and environmental factors. We highlight recent progress in key signaling pathways relevant to palatogenesis, their regulation by epigenetic mechanisms, and the newly appreciated role of non-coding RNAs in palatogenesis.

## 2. Craniofacial Development: Molecular and Genetic Basis

### 2.1. Anatomical Development and Classification of Cleft Lip and/or Palate (CL/P)

#### 2.1.1. Anatomical Overview of Palate Formation

The hard palate (bony part of the front) and soft palate (muscular part of the back) separate the oral and nasal cavities during embryogenesis. During palate development, shelves elevate, contact, and fuse at the midline to form the hard (anterior) and soft palates (posterior). This process is essential for both speech and swallowing. Palatal fusion begins at the back and progresses toward the front. Facial development begins early in pregnancy [1]. During development, the formation of the face and palate requires the spatiotemporal coordination of diverse cellular processes, including, but not limited to, growth, migration, differentiation, and apoptosis [2]. Palate development is a highly complex process that is regulated by transcription factors, growth factors, signaling molecules, and epigenetic regulators. An imbalance in fine-tuning can result in craniofacial defects such as CL/P. This process can be severely affected by environmental exposures, such as maternal smoking, addiction to medication, or environmental toxins, leading to the development of these congenital alterations [1].

The upper lip, philtrum, and primary palate develop from the fusion of the medial nasal and maxillary processes (Figure 1A) [3]; however, disruption of these processes can result in CL/P. The secondary palate fuses with the primary palate in its front and nasal septum in its anterodorsal region, both developing simultaneously (Figure 1B–D). This fusion forms a complete palate in the oral cavity, and separates it from the nasal cavity (Figure 1E). Palatal shelf elevation, contact, or fusion failure results in secondary cleft palate (CP). In humans, palatal development commences at approximately week 6 of gestation and is completed by week 12 [1]. In mice, this process starts at approximately E11.5, and is essentially completed by E15.5 (Figure 1) [2,3]. The complex process of palate formation depends on the precise spatiotemporal regulation of multiple factors, such as transcription factors, growth factors, signaling molecules, and epigenetic factors, for normal development. Disruption of this process owing to maternal smoking, medication abuse, or exposure to environmental factors can result in craniofacial deformities with CL/P. All of these are taken into an intricate balance to form and fuse the palatal shelves correctly. Slight disturbance of this fine balance leads to developmental anomalies, often CP, and other craniofacial abnormalities. Understanding the molecular mechanisms involved in palate formation is essential for developing effective preventive strategies and treatments for OFC disorders [3].

#### 2.1.2. Classification of Human Cleft Lip and/or Palate

A CL/P is a common birth defect that is distinguished by the location and extent of the CP. A CL occurs when the tissues of the upper lip do not fuse properly during fetal development, resulting in a gap or opening on one (unilateral) or both sides (bilateral) (Figure 2) [1].

There are three subtypes of unilateral cleft lip (CL).

-Incomplete CL (smaller gap);-Complete CL (full width of upper lip);-Median CL (rarely middle upper lip).

Bilateral CL is less common and more difficult to treat due to the severity of the deformity.

A CP is a deformity of the roof of the mouth caused by inappropriate fusion of the maxillary palatal processes during the early stages of pregnancy, resulting in either a complete (hard or soft) or incomplete palate (Figure 2).

-Complete CP involves both hard and soft palates.-An incomplete CP encompasses both hard and soft palates.-Submucous CP involves a small opening in the soft palate, with the mucous membrane remaining intact [1].

A submucosal cleft involves a soft palate defect; however, the overlying mucosal layer is intact. Diagnosis may be delayed until speech or hearing issues arise. The cleft lip and palate (CLP) frequently occur together (Figure 2) [1].

Asymmetric clefts affecting one or both sides of the lip or palate may affect classification and management [4]. Although evidence from mouse models of OFC asymmetry is limited, craniofacial structure formation can be regulated by genetic, epigenetic, and environmental factors similar to those in humans. Although many genetic causes and mutations associated with OFCs have been identified, gaps in knowledge regarding the cellular and molecular mechanisms involved mean that clinical care and even prevention strategies have not changed significantly [5]. Further studies using mouse models are essential to develop effective treatments.

## 3. The Pathogenesis of Orofacial Clefts in Humans Involves Genetic and Environmental Factors

### 3.1. Overview of Syndromic/Non-Syndromic Associated with Cleft Lip and/or Palate

CL/P is the most common congenital craniofacial malformation [6]. The prevalence of CL is approximately 1 in every 700 live births (Table 1) [1,6]. In the entire population, there are significantly more males than females born with CL [7]. The variation in occurrence shows a wide disparity among various racial and ethnic groups, with the African American population having a lower incidence (approximately 0.5/1000) [7]. A CL contains one or more clefts that extend from the upper lip to one or both nostrils. A CP is a type of fissure that forms on the roof of the mouth, upper lip, or both when the bones do not properly fuse during embryogenesis. This may affect one side, both sides or the central region (Figure 2) [1]. Although CL/P is not fatal, CL/P-affected patients suffer from dental, occlusal, functional, and aesthetic problems, along with secondary complications such as auditory, respiratory, and nutritional problems. The etiology of CP is multifactorial, and involves both genetic and environmental components [1]. Mutations in various genes, including monogenic disorders and chromosomal rearrangements, have been observed in patients with CL/P and are present in numerous genetic syndromes (Table 1 and Table 2) [8].

**Table 1 ijms-26-01382-t001:** Comprehensive genetic information of non-syndromic orofacial clefts: Prevalence, subtypes, and associated factors.

Feature	Description	Associated Genes/Regions/Pathways	References
Prevalence	~70% of all clefts; ~1 in 700 live births globally	N/A (not applicable)	[1,6,7]
Subtypes	45% cleft palate alone; 85% nonsyndromic cleft lip with or without palate	N/A (not applicable)	[8,9]
Genetic Factors	SNPs, single gene mutations, microRNA patterns	WNT pathway (*AXIN1*, *WNT9B*), Fgf10/Fgfr2/Shh pathway (*FGFR1*, *FGF2*), *COL11A1*, *IRF6*, *EGF*, *MSX1*, *PTCH*, *TGFB1*, *ROR2*, *FOXE1*, *TGFB3*, *RARA*, *APOC2*, *BCL3*, *PVRL2*	[9,10]
Chromosomal Regions	Linkage to >20 regions	chr 1p, 1q21, 1q32-42.3, 6p, 2p, 4q, 17q	[9]
Epigenetic Factors	DNA methylation	EWASs identify differentially methylated regions	[11]
Rare Variants	Mutations in specific genes	*ABCB1*, *ALKBH8*, *CENPF*, *CSAD*, *EXPH5*, *PDZD8*, *SLC16A9*, *TTC28* (*ABCB1*, *TTC28*, and *PDZD8* show significant mutation constraint)	[11,12]

**Table 2 ijms-26-01382-t002:** Genetic syndromes associated with orofacial clefts and their key features.

Syndrome	Key Features (Including OFCs)	Associated Genes /Chromosomal Regions	References
22q11.2 microdeletion syndrome	Cleft palate (most frequent), cardiac defects, immune deficiency, characteristic facial features	*TBX1* (within the 22q11.2 deletion), *miR-96-5p*	[13,14,15]
Van der Woude Syndrome (VWS)	Cleft lip, cleft palate, hypodontia, paramedian lower lip pits	*IRF6* (most common), *GRHL3*, *CDH2* (SNP rs539075), *NOL4*	[14,16,17,18]
Stickler Syndrome (STL)	Cleft palate/uvula, myopia, retinal detachment, joint problems, hearing loss	*COL2A1* (*STL1*), *COL11A1* (*STL2*), *COL11A2* (*STL3*), *COL9A1*, *COL9A2*, *COL9A3*, *LRP2*, *LOXL3*	[19,20,21]
Pierre Robin Sequence (PRS)	Micrognathia, glossoptosis (posterior displacement of the tongue), cleft palate, airway obstruction	*SOX9*, *BMPR1B*, deletions on 2q and 4p, duplications on 3p, 3q, 7q, 8q, 10p, 14q, 16p, and 22q	[16,22,23,24]
Kabuki Syndrome	Distinct facial features (midfacial hypoplasia, broad nasal tip, elongated palpebral fissures, large ears), cleft palate/high-arched palate, growth retardation, intellectual disability, congenital heart defects	*KMT2D * (most common), *KDM6A*	[25,26,27]
Wolf–Hirschhorn Syndrome (WHS)	Intellectual disability, growth delays, heart and skeletal defects, seizures, cleft palate, facial asymmetry	rearrangement t(4p;8p), t(4p;7p), t(4p;11p), t(4p;20q), t(4p;21q), and t(4p;12p)	[16,28,29]
CHARGE Syndrome	Coloboma, heart defects, atresia of the choanae, retarded growth/development, genital abnormalities, ear abnormalities/hearing loss, cleft palate	*CHD7*	[16,30]
Apert Syndrome (AS)	Craniosynostosis, midface hypoplasia, cleft palate (more commonly soft palate), syndactyly of hands and feet	*FGFR2 * (p.Ser252Trp, p.Pro253Arg)	[31,32,33,34,35,36]

With or without CP, CLs are normally divided into isolated (non-syndromic) and Mendelian syndromic forms. Non-syndromic cleft lip and/or palate (NSCLP) (93–97% of cases) has a complex etiology that is attributed to the interaction of genes with environmental factors (Table 1). The recurrence risk of NSCLP is estimated to be 4–10. Syndromic forms of CL/P account for approximately 5–7% of cases, encompass more than 200 different conditions, and have been recognized by varying patterns and prevalence of congenital malformations [9].

### 3.2. The Genetic and Epigenetic Basis of Craniofacial Abnormalities: Non-Syndromic and Syndromic Forms

#### 3.2.1. Non-Syndromic Craniofacial Anomalies

Non-syndromic orofacial clefts (NSOFC) account for 70% of all clefts, and are among humans’ most common birth defects. Its global prevalence is approximately 1 in 700 live births (Table 1) [8]. Non-syndromic clefts, defined as isolated deformities, comprise approximately 45% of CP alone and 85% of NSCLP (Table 1) [9]. NSCLP is associated with distinct genetic factors such as single nucleotide polymorphisms (SNPs), single gene mutations, environmental factors, and microRNA (miRNA) patterns (Table 1) [9]. The WNT pathway is critical for craniofacial development, and WNT pathway genes, including AXIN1 and WNT9B, have been associated with NSOFC [9,10]. Genes in this pathway, specifically FGFR1 and FGF2, have been associated with NSOFC [9]. Other genes associated with NSOFCs via linkage studies include *COL11A1*, *IRF6*, *EGF*, *MSX1*, *PTCH*, *TGFB1*, *ROR2*, *FOXE1*, *TGFB3*, *RARA*, *APOC2*, *BCL3*, and *PVRL2* [9]. Linkage analysis revealed that more than 20 chromosomal regions were linked to the NSOFC. Notable examples include chr 1p, 1q21, 1q32-42.3, 6p, 2p, 4q, and 17q [9].

Several epigenetic-wide association studies (EWAS) have been conducted on OFCs (Table 1). A study in the UK conducted EWAS using blood and lip tissues to test the association between methylation at each site and cleft subtype (cleft lip only (CLO) *n* = 50; cleft palate only (CPO) *n* = 50; CLP *n* = 50) [11]. Recently, ref. [12] demonstrated that eight genes (*ABCB1*, *ALKBH8*, *CENPF*, *CSAD*, *EXPH5*, *PDZD8*, *SLC16A9*, and *TTC28*) were consistently expressed in relevant mouse and human craniofacial tissues during facial formation and three genes (*ABCB1*, *TTC28*, and *PDZD8*) showed statistically significant mutation constraints. These findings highlight the role of rare variants in identifying candidate genes for NSOFCs.

#### 3.2.2. Syndromic Craniofacial Anomalies

While syndromic OFCs are more often attributable to one congenital cause or disrupted gene than NSOFCs, other challenges arise in determining the underlying mechanisms [9,16]. The underlying etiology associated with the OFC phenotype may be more difficult to determine for many syndromic conditions in which clefts are mild or uncommon. Often, multiple disease-causing genes and factors can exist, and the role of specific genes in cleft-associated cases may not be described thoroughly, particularly if clefts are a minor feature and not the primary focus of studies under particular conditions [16]. In addition, many syndromes are caused by deletions that disrupt several genes, further complicating the association between specific loci and lip and/or palate fusion (Table 2) [16].

##### 22q11.2 Microdeletion Syndrome

22q11.2 microdeletion syndrome is a congenital disorder with a broad phenotypic presentation that results predominantly from the microdeletion of chromosome 22 at a location known as 22q11.2 [13]. More than half of patients with 22q11.2DS (DiGeorge syndrome/velo-cardio-facial syndrome) exhibit craniofacial malformations, of which CP is the most frequently observed defect. Many of the developmental anomalies observed with this syndrome could be attributed to a decrease in copy number of genes located within the deleted region at 22q11.2, including the possible aberrant expression of the T-box TF (*TBX1*), whose role in palatogenesis has been well documented [14]. The loss and gain of *Tbx1* function suggests that *Tbx1* dosage is a critical determinant of normal palatogenesis. Tbx1 regulates craniofacial development through *miR-96-5p*, which represses *Tbx1* expression by binding to the 3′-UTR of its mRNA [14,15].

##### Van der Woude Syndrome

Van der Woude syndrome (VWS) is the most common form of syndromic clefting, accounting for approximately 2% of all CL/P cases, with a prevalence of 1/34,000 live births. VWS is an autosomal dominant disorder in which affected individuals have one or more of the following manifestations: CL, CP, hypodontia, or paramedian lower lip pits [16]. In this review, analysis of the VWS family identified an SNP, rs539075, located within intron 2 of the cadherin gene (*CDH2*) that may be associated with CL/P [17]. This group also identified an intronic variant of *NOL4* in patients with VWS, which co-segregated with CL/P [14].

##### Stickler Syndrome

Stickler (STL) syndrome is a heterogeneous disease characterized by collagen abnormalities (particularly collagen types II, IX, and XI). The prevalence of Stickler syndrome is estimated to be 1:7500–9000 [19]. STL is a disorder that includes congenital myopia, possibly coexisting with cataracts, retinal damage, cleft uvula, submucous cleft palate (SMCP), changes in craniofacial structure, changes in joints and bones, joint hypermobility, and progressive hearing loss [19].

STL is molecularly diagnosed based on the presence of pathogenic variants in six collagen-type genes (*COL2A1*, *COL11A1*, *COL11A2*, *COL9A1*, *COL9A2*, and *COL9A3*) and two non-collagen genes (*LRP2* and *LOXL3*) [20,21], following a predominantly autosomal dominant inheritance pattern. Mutations in *COL2A1* located on chromosome 12 (12q13.11) can cause SLT1 mutations [19]. STL2 occurs due to mutations in the gene encoding the α1 chain of collagen XI in *COL11A1*, which is located on the short arm of chromosome 1 (1p21.1). The main cause of Stickler syndrome type III (STL3) and non-ocular Stickler syndrome is deletions of the *COL11A2*-located on chromosome 6 (6p21.3), which encodes the α2 chain of collagen type XI [20].

##### Pierre Robin Syndrome

The Pierre Robin syndrome (PRS) refers to a set of characteristic craniofacial phenotypes commonly observed: glossoptosis, CP, micrognathia, and upper airway obstruction [16]. The Pierre Robin syndrome occurs in 1/8500–1/14,000 births [22]. Cleft palate is associated with deletions in 2q and 4p, and duplications in 3p, 3q, 7q, 8q, 10p, 14q, 16p, and 22q [22]. The dominant model suggests that mandibular hypoplasia causes a highly retropositioned tongue, blocking palatal shelf elevation and the airway. Alternatively, intrauterine mandibular compression or delayed neuromuscular development may restrict mandibular and tongue growth, leading to palate and airway obstruction [22]. Isolated PRS has been associated with mutations in or near the SRY-related HMG box 9 (*SOX9*) gene [23]. Mutations in *BMPR1B* have also been reported to cause PRS in two unrelated families [24]. Several genes encoding ECM components and ECM-interacting proteins have also been associated with syndromes, including clefts [16].

##### Kabuki Syndrome

Kabuki syndrome is a genetic disorder primarily characterized by distinct facial features, including midfacial hypoplasia, broad nasal tip, elongated palpebral fissures, and large abnormal earlobes. Other features include cleft or a high-arched palate, growth retardation, cognitive disabilities, and congenital heart defects [25,26]. Kabuki syndrome is caused by mutations in the *KMT2D* gene, which encodes an H3K4 histone methylase that promotes active gene transcription, or in the *KDM6A* gene, an X-linked histone H3K27 demethylase. Approximately 60–70% of cases are attributed to *KMT2D* mutations [25]. This mutated gene was the first pathogenic gene recognized in Kabuki syndrome, and is also known as *MLL2* [27]. Over 50 mutations have been identified at different sites of *KMT2D*, including nonsense, missense, frameshift, small deletions, and splice-site variants [26].

##### Wolf–Hirschhorn Syndrome

Wolf–Hirschhorn Syndrome (WHS) is a developmental disorder characterized by intellectual disability, growth delays, heart and skeletal defects, seizures, and sometimes midline issues, such as CP and facial asymmetry [16]. WHS is a rare contiguous gene deletion syndrome (prevalence of 1:20,000–50,000 births, with a female-to-male ratio of 2:1) induced by the absence of the distal portion of the short arm of chromosome 4 [28]. WHS typically arises from deletions in chromosome 4p16.3, which vary in size and position. Genetic defects are usually partial deletions of the distal short arm of chromosome 4, but the WHS phenotype can also be generated by complex chromosomal rearrangements, such as translocations or ring chromosomes [28]. Unbalanced translocations can be de novo or inherited from a parent with balanced rearrangement. The most frequently observed translocations are (1) those involving a rearrangement t(4p;8p), t(4p;7p), t(4p;11p), t(4p;20q), t(4p;21q), and t(4p;12p); (2) inverted duplications associated with terminal deletions in the same 4p arm; or (3) unbalanced pericentric inversions [28,29].

##### CHARGE Syndrome

Coloboma, heart, atresia of the choanae, retarded growth and development, genital abnormalities, ear abnormalities, and hearing loss (CHARGE) syndrome is a complex genetic disorder that affects multiple systems of the body. CHARGE syndrome is a multiple congenital malformation syndrome with an estimated birth prevalence of 1 in 15,000–17,000 newborns [16]. Mutations in chromatin remodeling and the gene expression regulator *CHD7* account for most cases. *CHD7*, located on chromosome 8, is responsible for CHARGE syndrome and was first discovered in a study that uncovered mutations in *CHD7* in individuals with this disorder [30]. This leads to a protein that plays a role in chromatin remodeling, both of which are important for gene expression during developmental processes. Disruption of embryonic development due to mutations in *CHD7* results in a complex phenotype involving various organs [30].

##### Apert Syndrome

Apert syndrome (AS) is one of the most common craniosynostosis syndromes worldwide. The prevalence of AS is 1/65,000 in the general population [31]. Patients with AS may develop oral problems such as severe maxillary hypoplasia, CL, and CP [32]. The CP occurs in approximately 30% of patients with Apert syndrome, with soft palate clefts being more common than hard palate clefts [33]. AS follows dominant genetic patterns and most patients are de novo cases caused by mutations in *FGFR2* [34]. *FGFR2* is the receptor for fibroblast growth factor (FGF), and is encoded by a gene at locus 10q26. FGFR2 is activated by binding to FGF and plays a role in cell proliferation, angiogenesis, and bone differentiation [35]. Mutation of exon IIIa of *FGFR2* can cause AS because it leads to an increased bone differentiation rate of Mesenchymal Stem Cells (MSCs) and the development of craniosynostosis. Common types of gene mutations are *FGFR2* p.Ser252Trp (S252W) of 755C > G and p.Pro253Arg (P253R) of 758C > G. The S252W mutation of *FGFR2* is usually accompanied by severe skeletal malformations of the craniofacial region and a higher incidence of CP, but the P253R mutation of *FGFR2* is often accompanied by more prominent syndactyly of the hands and feet [36].

### 3.3. Key Genes Involved in Craniofacial Development

#### 3.3.1. Morphological and Molecular Control of Palatal Shelf Growth and Patterning

In humans, lip closure and palatal fusion occur at 6 and 12 weeks gestation [37]. Because of these precise timings, animal models, particularly mice, are essential for studying craniofacial development [2,3]. Mice are key model organisms for investigating cleft lip and palate because they are genetically similar to humans and share similar facial developmental processes [3].

Palatal shelves are mostly composed of neural crest-derived mesenchyme [38]. A thin layer of oral epithelium borders them with a distinct anterior–posterior (A-P) axis. In mice, the embryonic development of the palate begins at approximately E11.5 (Figure 1A). Although neural crest cells (NCCs) begin to migrate to established positions, the palate, frontonasal projections, and palatal shelves have not yet been developed. E12.5 shows no obvious formation of frontonasal projections or palatal shelves (Figure 3A,B). By E13.5, the palatal shelves and vertical tissue plates of the palate began to increase and move toward each other (Figure 1B and Figure 3C,D). They eventually fuse along the midline between E13.5 and E15.5 (Figure 1C–E and Figure 4) [2]. The growth and fusion of these shelves are regulated by interactions between epithelial and mesenchymal tissues along the anterior–posterior axis [39,40,41].

SHH signaling is crucial for palatal shelf outgrowth, as it regulates cell proliferation and promotes the development of the palate [40]. SHH interacts with other signaling pathways, such as FGF and BMP, to ensure proper palatogenesis. The inactivation of *Shh* in the epithelium or mesodermal-specific inactivation of smoothened (*Smo*) can impair palatal cell proliferation and growth (Figure 3E) [42]. In addition, mice with co-mutations in *Hhat* and *Ptch1* show *Shh* gradient disruption during frontonasal protrusion development, resulting in hypoplasia of the central and lateral protrusions, ultimately resulting in CL and residual midline epithelial junctions [43]. Primary cilia are small hair-like projections extending from various tissues’ surfaces [44]. Essential for transmitting *Shh* signaling. Reduced expression of forkhead box F1 (*Foxf1*) in the palatal mesenchyme suggests that primary cilia are downstream effectors of *Shh* signaling (Figure 3E) [45].

This is essential for palatal shelf outgrowth, with its absence leading to a CP owing to impaired proliferation [46]. Although *Fgf10* is expressed in the mesenchyme, its receptor, *Fgfr2b*, is crucial in the epithelium, and epithelial-specific deletion of *Fgfr2* causes CP (Figure 3E) [47]. *Shh* signaling, which is dependent on *Fgf10*, is reduced in *Fgf10*^−/−^ and *Fgfr2b*^−/−^ embryos, highlighting a positive feedback loop between *Shh* and FGF signaling that regulates palatal proliferation [46,48]. These two signaling pathways and transcription factors work together to activate mesenchymal signaling to ensure proper palatogenesis and the establishment of the oral and nasal cavities. *Fgf10* maintains *Shh* expression in the palatal epithelium, whereas *Fgf7* suppresses *Shh* expression that is regulated by *Dlx5* (Figure 3E) [49]. Recent studies have demonstrated a complex regulatory network involving *Shh*, *Foxf1/2*, and *Fgf18* in developing the palatal shelves (Figure 3E). The ablation of *Foxf1* and *Foxf2* in mouse embryos interferes with palatal outgrowth, which affects the expression of *Fgf18* and *Shh* [41]. This coordination between transcription factors and FGF ligands, which is controlled by *Shh* signaling, regulates the growth and patterning of palatal shelves (Figure 3E).

Fgf9, a critical FGF ligand in craniofacial development, is expressed in the palatal epithelium and mesenchyme during palatogenesis in mice (Figure 3E) [50]. Additionally, *Sox11* null mice, with decreased *Fgf9* expression, exhibit an undersized mandible and CP, resembling CP caused by micrognathia and tongue malposition in the Pierre Robin Sequence (PRS) [51]. In a recent study, elevated *Fgf9* levels also induced TMJ dysplasia, impairing the spatial coordination between tongue descent and palatal shelf elevation, thereby exacerbating CP formation. TMJ dysplasia restricts the posterior dimension of the mandible and adds stress to the posterior palate, thereby increasing the likelihood of cleft formation. These findings suggest that TMJ dysplasia, which also co-occurs with CP in human syndromes such as achondroplasia and Muenke syndrome, may contribute to CP, even without reducing mandibular length [52].

Expression of the LIM homeobox genes *Lhx6* and *Lhx8* negatively regulates proliferation of the maxillary arch and palatal mesenchyme by repressing FOX family transcription factors and the cell cycle inhibitor Cdkn1c (p57Kip2) (Figure 3E) [53]. The maintenance of mitochondrial homeostasis by *Lhx6* is mediated through PINK1/Parkin-mediated mitophagy and the MAPK signaling pathway. The transcriptional downregulation of *Lhx6* by RA impairs the maintenance of mitochondrial homeostasis at the transcriptional level; hence, it causes defects in the proliferation and migration of HEPM cells and CP formation [54]. Although the *Shh*-*Foxf1/2*-*Fgf18*-*Shh* molecular circuit is known to be involved in early palatal development (Figure 3E), it is unclear whether *Lhx6/8* also influences the *Shh* and FGF signaling network during palatal shelf formation. Additionally, transforming growth factor-β (*Tgf-β*) signaling affects *Shh* signaling in the palatal mesenchyme by regulating lipid metabolism [55].

The Bone Morphogenetic Protein (*Bmp*) signaling pathway regulates cell proliferation, cell differentiation and apoptosis, which are critical steps in the morphogenesis of the face [56,57]. The BMP pathway may interact with other cellular pathways, such as the *Shh* signaling pathway, which plays a crucial role in craniofacial development [58], and interacts with the palatal mesenchyme, where loss of *Smo* results in increased *Bmp4* expression and decreased levels of *Bmp2* (Figure 3E) [48]. *Shh* signaling promotes the activity of *Bmp2* to stimulate cell proliferation in the palatal mesenchyme [59]. The role of *Bmp2* is during the development of the facial process in craniofacial morphogenesis, and the essential role of *Bmp4* is in tissue differentiation, along with the establishment of facial prominence [60]. Although complete inactivation of *Bmp4* is lethal during early embryonic stages, its targeted deletion in the maxillary mesenchyme and oral epithelium results in CL without affecting the secondary palate [56]. Overexpression of the BMP antagonist Noggin in the palatal mesenchyme causes delayed palatal growth and CP [61]. This highlights the essential role of *Bmp* signaling in normal palatogenesis, and dysregulation of this process results in CL or CP [62]. Studies have shown that *Bmpr1a*, a type I Bmp receptor, is essential for palate formation (Figure 3F) [56]. Deletion of *Bmpr1a* in the maxillary mesenchyme and oral epithelium of mice causes CLP, whereas its loss in the oral epithelium alone does not [63]. This suggests that *Bmpr1a* signaling in the mesenchyme is crucial for palatogenesis. Conditional deletion of *Bmpr1a* in NCCs leads to severe craniofacial defects [64], whereas its inactivation in the palatal mesenchyme results in anteriorly restricted CP and reduced cell proliferation. Loss of *Bmpr1a* also disrupts *Shh* expression, indicating that BMP-SHH interactions regulate palate growth [65]. Additionally, loss of the BMP antagonist Noggin causes CP with increased apoptosis and decreased cell proliferation [66], highlighting the need for tightly regulated *Bmp* signaling during palate development.

WNT signaling is crucial for *Pax9*-mediated secondary palate development [67,68,69], and regulates cell proliferation, migration, and differentiation [70]. In *Pax9*^−/−^ mice, decreased Axin2 and β-catenin levels and increased Dkk2 expression (Figure 3F,G) disrupted WNT signaling, but pharmacological inhibition of DKK partially rescued palate morphology. Inactivation of Sostdc1 restores WNT signaling and rescues CP (Figure 3G) [67]. Pathogenic variants in WNT pathway genes, such as *Wnt3a*, are linked to NSCLP [71]. WNT signaling disruptions can lead to CL/P and are also associated with other conditions [8], including cancer [72] and skeletal disorders [73].

In *Pax9*^−/−^ mice, EDA/EDAR signaling downstream of WNT signaling is reduced, but is not essential for palatogenesis [69]. In utero stimulation with EDAR agonists restored CP in these mice, and the creases appeared disorganized and did not influence the expression of *Bmp4*, *Msx1*, *Fgf10*, or *Osr2*. These studies indicate that *Pax9* acts through the WNT signaling pathway by regulating antagonists of WNT in the palatal mesenchyme (Figure 3E). However, further studies are needed to understand how *Pax9* regulates WNT target genes.

#### 3.3.2. Molecular Regulation and Regional Patterning Along the Anterior–Posterior Axis of Palatal Development

The developing palatal shelves are molecularly and morphologically regionalized along the A-P axis, where regions of the anterior express transcription factors different from those of the posterior parts [74]. *Msx1* and *Shox2* are required for anterior palatal mesenchymal cell proliferation (Figure 3F), whereas the posterior region expresses *Meox2* and *Tbx22* (Figure 3G). *Msx1* acts via *Bmp4* in the mesenchyme to regulate *Shh* expression in the anterior palatal epithelium (Figure 3F) [59], whereas *Mn1* and *Barx1* are expressed more posteriorly (Figure 3G) [75]. *Msx1* maintains *Shh* expression in the anterior palatal epithelium by regulating *Bmp4* in the mesenchyme (Figure 3F), whereas *Mn1* and *Barx1* are primarily expressed in the posterior region (Figure 3G). Mice with disrupted *Msx1* or *Mn1* genes exhibited complete CP, but the defects were region-specific. *Msx1*^−/−^ mice have proliferation defects only in the anterior palate, whereas *Mn1*^−/−^ mice have growth defects in the middle and posterior palates [59,75]. *Shox2*^−/−^ mice have a cleft restricted to the anterior palate, whereas the posterior palate develops normally, demonstrating the role of *Shox2* in anterior palatal expansion [76]. In contrast, *Tbx22*^−/−^ mice experience varying cleft severity, from full CP to SMCP, with *Tbx22* acting downstream of *Mn1* in posterior palatal outgrowth [75]. *Msx1* and *Shox2* expression in the anterior palate is regulated by BMP signaling, as evidenced by the decreased expression in *Wnt1*-Cre; *Bmpr1af*^−/−^ mice [64]. In palatal explant cultures, *Msx1* expression was specifically induced in the anterior palatal mesenchyme by *Bmp4* [74], while exogenous *Bmp4* could not stimulate *Shox2* expression. However, the anterior palatal epithelium was able to induce *Shox2* expression in the posterior mesenchyme, revealing distinct differences between the epithelium and mesenchyme along the anterior–posterior (A-P) axis (Figure 3F) [76]. Furthermore, canonical Wnt signaling is restricted to the anterior palatal mesenchyme and depends on Gpr177 for Wnt secretion. In particular, *Wnt5a* expression is high in the anterior mesenchyme. It regulates mesenchymal migration and elongation of the palatal shelf and its transcription is controlled by *Msx1* (Figure 3F) [77]. In particular, LIM domain transcription factors, along with the cofactor Ldb1, have been identified in palatal growth and patterning, and their chemical and genetic inactivation leads to posterior mesenchymal ectopic expression of *Wnt5a* (Figure 3F) [78]. These findings highlight the distinct molecular mechanisms involved in A-P patterning of the palate.

#### 3.3.3. Regulatory Networks and Patterning Along the Mediolateral Axis

SHH signaling plays a key role in palatal development because its disruption reduces its expression in the palatal mesenchyme [48]. The expression of *Osr2* is dependent on *Pax9*, and embryos lacking both *Osr2* and *Pax9* exhibit CP, along with decreased *Fgf10* expression in the palatal mesenchyme (Figure 3E). This suggests the importance of *Osr2* and *Pax9* in palatal development and regulating *Fgf10* levels [79]. Patterning along the mediolateral axis of the palate is critical for establishing the gene expression domains that provide proper growth and fusion. Around E12, the lateral side of the palatal shelves begins to form palatal rugae, and *Shh* expression is restricted to this region [45]. The zinc-finger transcription factors *Osr1* and *Osr2* exhibit graded expression along the mediolateral axis of the developing palatal mesenchyme (Figure 3E). By E13.5, *Osr1* expression was confined to the lateral side, whereas *Osr2* was strongly expressed in the lateral mesenchyme, tapering medially. Deletion of *Osr2* leads to CP because of reduced cell proliferation on the medial side and disrupted patterning. *Osr2* partly compensates for the role of *Osr1*, as evidenced by the repair of CP in *Osr2*-deficient mice with *Osr1* cDNA [80].

Increased expression of osteogenesis-related genes, such as *Mef2c*, *Sox6*, *Sp7*, and several BMP ligands (Bmp3, Bmp5, and Bmp7), as well as ectopic expression of class-3 Semaphorins (*Sema3a*, *Sema3d*, and *Sema3e*), was observed in *Osr2*^−/−^ mice. This highlights the role of *Osr2* in the repression of mesenchymal cell proliferation and the prevention of premature osteogenesis. The function of Semaphorins in palatogenesis is yet to be determined [81]. A *Dlx5*-dependent transcriptional pathway regulates mediolateral patterning and palatal expansion (Figure 3E). *Dlx5* is co-expressed in the medial mesenchyme of the palatal shelf with *Fgf7*; the expression of the latter gene is dramatically downregulated in the palates of *Dlx5* mutant embryos. This reduction in *Fgf7* expression may cause the expansion of *Shh* expression into the medial palatal epithelium, as exogenous *Fgf7* can inhibit *Shh* expression in palatal explant cultures. Although palatal shelves in *Dlx5*-deficient mice are elevated and fused, the oral palate is significantly enlarged and the soft palate deformed [2].

Interestingly, although *Msx1*-deficient mice show reduced expression of *Shh* in the anterior palate, compound mutants that lack both *Dlx5* and *Msx1* express *Shh* within the medial epithelium, compensating for cell proliferation defects caused by *Msx1* [49]. This study identified a new pathway involving *Dlx5* and *Fgf7* in the mediolateral patterning and palate growth. However, because *Fgf7*-deficient mice do not display overt palatal defects, another signaling molecule could act downstream of *Dlx5* to modulate *Shh* expression [1].

#### 3.3.4. Genetic Network Controlling Palatal Shelf Adhesion and Fusion

Concomitantly, the growth of the palatal shelves supports the development of the maxillary and mandibular processes; however, this only occurs because of downward and forward movement of the tongue. This is required to elevate the palatal shelves, which then contact and fuse the midline (Figure 1 and Figure 4B,C) [2]. An elaborate interaction of signaling pathways is engaged in the process that promotes the adhesion and fusion of shelves. Mesenchymal integrity in the fused palate is compromised by the removal of the mesenchyme between the shelves (Figure 4F). Disruption of midline marginal epithelial differentiation, adhesion capacity, and loss of mesenchymal–epithelial transition can lead to a CP. Mutations or dysfunction in genes such as *Jag2*, *Fgf10*, *Irf6*, and *Grhl3* result in inadequate adhesion or fusion of palatal shelves, leading to CP [37,82]. The absence of *Jag2*, a Notch ligand, causes CP in *Jag2z^ΔDSL/ΔDSL^* mice mainly because of abnormal adhesion of the palatal shelves to the tongue. *Jag2* is expressed in the oral epithelium and maintains periderm cells, which are essential for regulating fusion competence (Figure 4D) [37]. *Fgf10*^−/−^ embryos also showed reduced *Jag2* expression and palatal–tongue fusion defects, suggesting that *Fgf10* regulates palatal development upstream of Jag2-Notch signaling (Figure 4D). Mice with functional interferon regulatory factor 6 (*Irf6*) mutations due to homologous splicing null or R84C point mutations exhibit undifferentiated hyperproliferative epidermis, resulting in various developmental abnormalities including CP and inappropriate oral adhesions [82]. *Irf6* cooperates with *Jag2* to regulate epidermal differentiation as demonstrated by severe defects in *Irf6*R84C/+; *Jag2^ΔDSL/+^* mice [83]. This phenotype was similar to that observed in mice with homozygous *Irf6* or *Jag2* alleles, highlighting the importance of these genes in palatal development (Figure 4D). Expression of either gene is unaffected in individual mutants, indicating that *Irf6* does not directly regulate *Jag2* expression (Figure 4D) [82]. Mice lacking the p63 transcription factor show CP and undifferentiated epidermis [1], with reduced *Irf6* expression in the palatal epithelium [84]. Compound mutant mice, *p63*^+/−^; *Irf6*^R84C/+^, also exhibited failed palatal shelf fusion due to improper periderm cell maintenance. p63 may positively regulate *Jag2* and *Fgfr2* expression, although its relationship with the Jag2-Notch and Fgf10-Fgfr2b pathways in palatal epithelial differentiation is not fully understood [85]. The absence of Ikk-α or *Tbx1* in mouse embryos leads to abnormal oral adhesions between the tongue and palatal shelves, indicating that palatal epithelial differentiation is controlled by a genetic network that includes *Irf6*, *Jag2*, *p63*, *Ikk-α*, *Tbx1*, and *Fgf10-Fgfr2b* signaling pathways (Figure 4D) [86].

Periderm removal and disappearance of the medial edge of the palatal shelf are two important events in determining whether palatal fusion can occur, and abnormalities such as abnormal oral adhesions do not occur. However, the mechanisms controlling periderm removal and midline epithelial seam (MES) disappearance need to be determined. There are three dominating hypotheses on the way in which the MES disappears [87,88]. According to one hypothesis, this may involve epithelial–mesenchymal transition (EMT). EMT may enable the MES epithelium to integrate into the mesenchyme of the non-CP. Several in vivo lineage analyses have been performed using epithelial-restricted Cre-expressing transgenic lines and ROSA26R reporter lines to trace MES cell fate. For example, a study examining lacZ expression in *Shh*GFPCre or *K14*-Cre mice crossed with R26R reporter mice did not report the presence of lacZ-expressing mesenchymal cells. It thus concluded that EMT was not a major mechanism for MES regression [87]. In contrast, another study reported mesenchymal β-galactosidase activity in *K14*-Cre; R26R embryos before and during MES regression [89], which may be related to different Cre levels or expression patterns between the various *K14*-Cre transgenic mouse lines.

ESRP1 and its paralog ESRP2 are epithelial splicing regulatory proteins that co-localize with *Irf6* and function in the embryonic epithelium to regulate craniofacial development and EMT during embryogenesis (Figure 4D) [90]. Functional studies have shown that *Esrp1/2* mutations result in defective splicing of pre-mRNA which, in turn, causes aberrant isoforms of CTNND1, leading to weakened epithelial integrity and OFC anomalies. In addition, studies have identified ESRP1/2-controlled isoforms of CTNND1 that regulate epithelial adhesion and WNT signaling, implicating disrupted splicing in craniofacial anomalies (Figure 4D) [91].

Apoptosis plays a crucial role in the dissolution of the medial edge of the palatal shelf during palatal fusion, allowing the mesenchyme to become connected. The usual signs of apoptosis, including TUNEL positivity and active caspase 3, are commonly detected in MES cells during this process, with very few proliferating cells observed in this region [87,92]. However, recent research examining the role of the *Apaf1* gene, which is involved in caspase 3-mediated apoptosis, found that *Apaf1* deficiency does not affect palatal fusion or MES dissolution [93], contradicting earlier studies that reported fusion issues in *Apaf1*-deficient embryos [92]. This may be due to the fact that, in most of the earlier studies, the palate evaluation was incomplete. While apoptosis is one of the major mechanisms of MES breakdown, further studies are necessary to clarify the molecular mechanism of palate fusion, including Tgf-β signaling. Among these, *Tgf-β3*, exclusively expressed in the medial edge epithelium (MEE), plays an important role in the removal of MES (Figure 4E). The absence of Tgf-β3 in embryonic mice leads to improper midline contact between the palatal shelves and the persistence of MES [1].

*Tgf-β* signaling is essential for palatal fusion and is activated through type I and type II receptor dimers, leading to phosphorylation of R-Smads and transcriptional regulation. *Smad2* is crucial for MES breakdown and *Smad2* overexpression can partially restore fusion in *Tgf-β3* deficient mice. However, the deletion of Smad4 does not affect fusion, suggesting that other pathways, such as the p38 MAPK pathway, are involved [94]. *Tgf-β* signaling activates *Tak1*, which works independently of the Smad pathway, promoting palatal fusion through both *Smad* and p38 MAPK-dependent mechanisms [2]. *Irf6* regulates periderm differentiation and is activated in periderm and basal MEE cells before fusion (Figure 4E). Its absence in mutant embryos resulted in failed fusion; however, *Irf6* overexpression restored this process. *Irf6* downregulates *p63* and increases *p21* expression, facilitating cell cycle exit and MEE degeneration (Figure 4E) [95,96]. *Tgf-β3* downregulates *Jag2* in MEE, and blocking Notch signaling can partially restore palatal fusion in *Tgf-β3*-deficient cultures [97]. Oral periderm integrity is maintained by Jag2-Notch signaling [37]. A reduction in *Jag2* expression within the MEE is a key mechanism through which *Tgf-β3* disrupts periderm function and promotes palatal shelf adhesion. Beta-catenin (*Ctnnb1*) also plays a role in palatal fusion by regulating *Tgf-β3* expression in MEE. Destruction of β-catenin epithelial cells (*Ctnnb1*) results in decreased apoptotic MES cells in the MEE, loss of *Tgf-β3* expression, and failure of palatal shelf fusion and CP. However, β-catenin can function in adherent junctions or in the canonical Wnt signaling pathway [98], and the exact mechanism of its involvement in MES dissolution requires further research.

Several transcription factors are crucial for palatal fusion. The Snail family, including *Snai1* and *Snai2*, plays a key role, as fusion fails in *Snai1*^+/−^; *Snai2*^+/−^ compound mutants along with reduced MES apoptosis (Figure 4E). Interestingly, although *Tgfβ-3* expression remains unaffected in these mutants [99], exogenous *Tgfβ-3* can induce *Snai1* expression through a *Smad*-independent pathway, suggesting that Snail factors may act downstream or in parallel with *Tgfβ-3* signaling. *Runx1* is a transcription factor involved in palate development, and is expressed throughout the MEE during palatal fusion (Figure 4E) [1]. Disruption of *Runx1* results in anterior-specific failure of palatal shelf fusion and a cleft between the primary and secondary palates. This failure is linked to a distinct region in the anterior MEE, with less TUNEL staining and unique behavior [42]. In contrast, the *Meox2* transcription factor is crucial for maintaining the integrity of the posterior palate after fusion. *Meox2*^−/−^ embryos show a post-fusion split in the posterior palate [93].

*Irf6* is essential for *Snai2* expression in MEE cells, and *Snai2* knockdown slows palatal fusion in explant culture [100]. Reverse signaling of ephrin enhances *Snai1* expression in MEE cells and can partially rescue fusion in the presence of Tgf-β3-blocking antibodies, suggesting cooperation between ephrin and *Tgf-β3* signaling in regulating palatal fusion [101]. *Snai1* and *Snai2*, acting downstream of *Tgf-β3*, downregulate E-cadherin, which may loosen MEE and periderm cell adhesion, leading to periderm desquamation (Figure 4E). Given the critical role of *Irf6* in these processes, the MCS-9.7 regulatory region may act as a mutational hotspot for both rare and common genetic variations. These variations could lead to or increase the risk of different forms of OFCs by disrupting *Irf6* expression in the periderm or basal layers of the oral epithelium [102]. *Tgf-β3* and *Irf6* also induce MMP13, which is involved in basement membrane degradation in the MEE. CEACAM1, expressed in the periderm before fusion, is also involved in the process of palatal fusion because *Ceacam1*^−/−^ embryos present a delay in fusion, whereas its relationship with *Tgf-β3* signaling is unknown [103]. Further studies are needed to determine how desquamation, apoptosis, and *Tgf-β3*-mediated periderm cell death are interrelated. The TGF-β signaling pathway is important in several biological and cellular processes, including cell growth regulation, immune responses, and embryonic development [104]. *Shh* signaling modulates lipid metabolism in the palatal mesenchyme. In facial morphogenesis, *Tgf-β* signaling is essential for palatal fusion through its interaction with other signaling pathways, such as WNT, FGF, and BMP. *Tgf-β* is involved in EMT, which is a vital step for successful palatal shelf migration and fusion [105]. The *Tgf-β* signaling pathway involves several genes, and pathogenic variants in some of these genes have been shown to be associated with the development of OFCs, such as the variants *IRF6* gene associated with VWS [18], the *SMAD* gene family, which also cross-interacts with the BMP signaling pathway, and variants in these genes are associated with an increased risk of CL development [105]. The genetic process of palatal development involves both genetic and epigenetic factors, including miRNAs that regulate gene expression during palatal fusion.

### 3.4. Epigenetic Mechanisms Landscape in Palatogenesis: Molecular Dynamics and Developmental Regulation

#### 3.4.1. Overall Epigenetic Modifications in Development

Epigenetics refers to mechanisms that alter gene expression by modifying the chromatin structure rather than changing the DNA sequence itself [106]. DNA is tightly wrapped around histone proteins to form nucleosomes, which are basic units of chromatin [107]. The arrangement of chromatin determines whether DNA is transcriptionally active; loosely packed chromatin is typically active (euchromatin), while densely packed chromatin is generally inactive (heterochromatin) Epigenetic modifiers, which include “writers”, “erasers”, and “readers”, regulate chromatin structure through various mechanisms such as DNA and histone modifications, large protein complexes, and non-coding RNAs DNA methylation generally occurs in CpG islands near promoter regions, while histone modifications involve the addition of chemical marks to histone tails, influencing the recruitment of transcriptional machinery. Protein complexes, such as polycomb repressive complexes and chromatin remodeling complexes, alter chromatin architecture and DNA accessibility. Non-coding RNAs, including miRNAs and long non-coding RNAs, are involved in gene silencing and chromatin regulation (Figure 5) [108]. When these epigenetic regulators are disrupted by mutations, gene expression can become aberrantly activated or repressed, leading to diseases such as cancer and neural crest-related disorders. Epigenetic modifications provide additional genetic information that can be inherited across generations. DNA methylation and histone modification in mammals regulate gene expression and affect cell fate during development [109]. These dynamic modifications can vary during developmental processes such as craniofacial and neural tube development, tissue regeneration, and senescence [110]. DNA methylation and histone modifications also regulate genomic imprinting, which is the parent of the origin of certain genes. Epigenetic modifications are influenced by environmental factors, such as folates and retinoids, which contribute to altered developmental outcomes, including NSOFC, CLP, and CPO [111]. Epigenetic modifications may explain variations in OFC prevalence across populations, which cannot be attributed solely to genetic differences [112].

#### 3.4.2. DNA Methylation Dynamics in Palatogenesis and Craniofacial Development

DNA methylation involves adding a methyl group to cytosine nucleotides in CpG sequences [113], often in CpG islands within promoter regions. This process recruits transcriptional repressors that inhibit gene expression by blocking transcription factors (Figure 5) [114]. In mammals and other vertebrates, methylation of cytosine (C) at the C5 position, leading to the formation of 5-methylcytosine (5mC), is widely accepted as the only epigenetic form of DNA methylation [8]. The methylation of adenine (A) in vertebrates remains controversial, whereas the methylation of guanine (G) and thymine (T) has not been reported. DNA methylation is catalyzed by DNA methyltransferases (DNMTs), which use S-adenosylmethionine (SAM) as an exclusive donor for methyl groups. Thus, the activity of DNMTs (histone-modifying enzymes) depends on an adequate supply of SAM, which is produced through folate and methionine cycles [112].

Methylation is thought to promote gene silencing by preventing the binding of transcription machinery or activators to DNA through spatial interference. In addition, methyl-CpG binding proteins recruited to 5-methylcytosine (5mC) can activate transcriptional repressors such as histone deacetylases [115]. Methylation generally occurs at cis-regulatory elements, particularly promoters and enhancers, to control variability in gene expression [116]. Promoters adjacent to the 5′-untranslated region (5′-UTR) are where the transcription machinery binds and initiates transcription. Enhancers located near the promoter or in distant regions, including the 3′-untranslated region (3′-UTR), interact with transcription factors to promote gene expression. They can physically form loops interacting with promoters and promoting transcriptional activation [117]. Enhancer activity is highly tissue-specific and plays a key role in spatial and temporal gene expression dynamics during embryonic development [118]. Although many studies have focused on promoter methylation [116], recent evidence suggests that enhancer and gene body methylation may play equal or even greater roles in regulating gene expression during development [119]. Methylation occurs at specific base sequences or motifs, and regulates gene expression across generations. Epigenetic methylation is commonly found in regions enriched with 5′-CpG-3′ motifs known as CpG islands [116]. Methylation at CpG islands is symmetrically maintained on both strands, making it a heritable epigenetic marker that does not require de novo methylation for reestablishment. Promoters are strongly linked to CpG islands, whereas enhancers may or may not be associated with them. Because of the correlation between promoters and CpG islands, early studies primarily focused on promoter methylation as the major epigenetic control of gene expression regulation. Another significant epigenetic marker is the pentanucleotide motif, 5′-CCWGG-3′ (where W can be A or T), which undergoes methylation inside C. Methylation of 5′-CWG-3′ has been conserved across generations in mammalian cells, likely because of its recognition as 5′-CCWGG-3′ by methyltransferases. The stable methylation of this motif challenges the previous notion that only CpG islands contain C bases suitable for transgenerational marking. The 5′-CCWGG-3′ motif is crucial for enhancer methylation dynamics in orofacial development [8] and is gaining increasing attention.

Adenine methylation, specifically the conversion of adenine nucleobases into N6-methyldeoxyadenine (N6mA), is a known modification of RNA that plays the role of N6-methyladenosine (m6A) in mammalian RNA processing [120]. However, their roles in mammalian DNA methylation remain controversial. Although N6mA is a well-established modification in prokaryotes and some eukaryotes, such as *Caenorhabditis elegans*, where it influences mitochondrial stress adaptation [121], its presence and function in mammalian DNA is disputed. Recent studies have questioned earlier claims regarding N6mA as a functional epigenetic marker of mammalian DNA. This suggests that the supposed evidence of N6mA in mammalian DNA may be due to RNA contamination or technical issues [122]. N6mA may be misincorporated into DNA by polymerase activity during the processing of ribo-N6mA, rather than functioning as a true epigenetic marker [123]. In RNA, N6mA modifications have been well established, especially in the development context. The potential role of N6mA in RNA during orofacial development was also discussed. RNA methylation, often mediated by *Nsun* family genes [112], is highly expressed in mouse embryonic tissues involved in craniofacial development, suggesting that RNA modifications may play a role in the etiology of OFCs.

DNA methylation is catalyzed by a family of DNA methyltransferases (DNMTs) [124], including DNMT1, DNMT3A, and DNMT3B, which play important roles in cell fate determination and tissue specification by modulating gene expression. DNMT1 is a methyltransferase that maintains methylation patterns through DNA replication and repair. The de novo methylation functions, on the other hand, are provided by DNMT3A and DNMT3B, which stablish new methylation marks mainly at the promoter region. These enzymes play significant roles in regulating gene expression, especially early in development and in tissue-specific GRNs. In mice, *Dnmt3a* and *Dnmt3b* are highly expressed in undifferentiated embryonic stem cells, and their expression decreases as cells differentiate [106]. DNMT3A represses neural genes such as *Sox2* and *Sox3* to promote neural crest specification during chicken development [125]. Zebrafish Dnmt3b and methyltransferase G9a regulate neurogenesis and craniofacial skeletal element formation [126]. Studies in human embryonic stem cells have shown that knockdown of *DNMT3B* accelerates neural and neural crest differentiation by upregulating neural crest specification genes such as *PAX3*, *PAX7*, *FOXD3*, *SOX10*, and *SNAIL2* [127].

Studies have shown that DNMT3B is essential for neural crest and craniofacial development; however, conditional loss of *Dnmt3b* in mouse NCCs results in only mild neural crest migration defects and no significant craniofacial phenotypes [128]. This suggests that DNMT3B may function earlier in development than previously thought or may affect other tissues that are secondary to neural crest development. Recent studies have shown that DNMT3B can function without a catalytic domain and may act as a secondary cofactor to support the enzymatic activity of other DNMTs, such as DNMT3A [129]. Future studies should explore the dual role of DNMT3B better to understand its role in craniofacial and neural crest development.

Several studies using mouse models have explored the role of differential gene methylation in orofacial development, particularly in palatogenesis. CpG methylation in the palate is significantly higher at embryonic day E14.5 than at E13.5 and E18.5, a critical time when the palatal shelves are elevated above the tongue, just before medial epithelial seam formation [8]. A microarray approach was used to examine DNA methylation in mouse palates from E12 to E14. They found that 73% of the detected genes were methylated, mostly within gene bodies rather than promoters, with 30% of methylation occurring in CpG islands [130]. These findings align with those of previous studies on RA exposure, where differentially methylated regions (DMRs) were located at intronic enhancers of genes linked to palatogenesis. *Sox4*, a key gene in palatal development, showed decreased expression at E13 and E14 because of methylation in the CpG-poor promoter region. *Sox4* plays a role in integrating several signaling pathways, including Tgf-β, Wnt/β-catenin, BMP, FGF, and Hedgehog, which regulate palatal fusion and extension [8].

Mutant mouse strains have also shed light on the role of methylation in orofacial development. The A/WySn strain, which has a 15–20% risk of CL/P, involves an epistatic interaction between *Clf1* (an IAP retrotransposon at the 3′ end of Wnt9b), *Clf2*, and a maternal effect. *Clf2* suppresses *Clf1* IAP via DNA methylation. *Clf1* was later identified as a metastable epiallele with stochastic methylation during embryogenesis, indicating that some individuals lack *Clf1* methylation, leaving them vulnerable to CL/P [131].

#### 3.4.3. Epigenome-Wide Association Studies (EWAS) in Orofacial Clefts (OFCs)

EWAS have identified significant DNA methylation differences linked to OFCs, particularly NSCLP. For example, a study in the UK found differentially methylated regions in the blood and lip tissue across cleft subtypes, including well-known cleft-related genes *TBX1*, *COL11A2*, *HOXA2*, and *PDGFRA*, and identified 250 new loci [11]. A later study conducted in Brazil revealed 578 methylation variable positions associated with NSCLP that were highly enriched for regulatory regions involved in craniofacial development [132]. Long interspersed nucleotide element-1 (LINE-1), a marker of global DNA methylation, was found to be differentially methylated in NSOFCs compared to controls [133]. Mutations I n the 5,10-methylenetetrahydrofolate reductase (MTHFR) gene, such as c.C677T and cA1298C, have been reported to reduce DNA methylation levels [9], whereas increased methylation levels of LINE-1 have been observed in the center of CL in the c.C677T mutation [121]. These DMRs may explain the absence of hereditary patterns in CL. With environmental pressures, population-specific epigenetic modifications can be anticipated, and worldwide investigation of cleft populations is needed to understand their epigenetic contribution to CP etiology.

In the last decade, methylation profiling has identified epigenetic modifications as key players in the etiology of OFCs. These modifications are particularly appealing as mechanisms for the environmental causes of OFCs. For example, maternal smoking has been shown to differentially methylate genes previously associated with OFCs in children, including *MSX1*, *PDGFRA*, *GRHL3*, *ZIC2*, and *HOXA2* [134]. Other methylation profiling studies have identified genes with variable methylation levels that may influence the incidence of OFCs. These include transcription factors (*LHX8*, *PRDM16*, *PBX1*, *GSC*, *VAX1*, and *MYC*), growth factors and modulators (*WNT9B*, *BMP4*, *EPHB2*, *BICC1*, and *DHRS2*), extracellular matrix genes (*CRISPLD2*, *NTN1*, and *CDH1*), and miRNAs (*MIR140* and *MIR300*) [132]. Some of these genes, including *PRDM16*, *BHMT2*, and *WHSC1*, encode proteins involved in methyltransferase activity [135]. Further studies have explored the variable methylation positions across different OFC subtypes and identified hundreds of methylation variable positions distinguishing between CLP, CLO, and CPO [11]. These findings suggest that DNA methylation profiling is a promising approach that could offer a more detailed understanding of the etiology of OFC.

#### 3.4.4. Impacts of Histone Modifications in Craniofacial Development

Histone modifications, the chemical modifications of histone proteins, are complex in nature and important in regulating chromatin structure and gene expression, especially development (Figure 5). Specifically, PTMs, such as methylation, acetylation, deacetylation, phosphorylation, ubiquitination, and sumoylation dynamically modulate gene expression by compaction and relaxation of chromatin [108], and therefore chromatin accessibility. These changes are tightly linked to cell fate decisions during NCC development, and regulation of these alterations is especially critical [136]. Another essential PTM that affects chromatin accessibility and gene expression is histone acetylation, which is regulated by histone acetyltransferases (HATs) and histone deacetylases (HDACs) (Figure 5).

NCC migration and differentiation also rely on HDAC function, and mutations in *HDAC1*, *HDAC2*, and *HDAC4* have been linked to craniofacial defects [137]. In addition, NCCs depend on specific histone modifications, such as H3K4me1 and H3K27ac, in enhancer regions to regulate chromatin structure and gene expression during development [138]. DNA methylation and histone modifications often work together, and environmental factors such as 2,3,7,8-tetrachlorodibenzo-p-dioxin (TCDD) exposure can disrupt histone acetylation, thereby affecting developmental processes such as CP formation in mice [139]. Mutations in histone-modifying enzyme genes are commonly associated with developmental disorders, such as OFC [16]. Taken together, the functions of enzymes such as histone methyltransferases, demethylases, HATs, and HDACs are crucial for the proper development of the neural crest. The disruption of these processes can affect NCC proliferation, migration, and differentiation, resulting in congenital craniofacial birth defects.

##### Histone H3K27me3 Demethylase KDM6A, KDM6B

Kabuki syndrome, caused by mutations in *KDM6A* or *KMT2D*, leads to developmental abnormalities. KDM6A, an X-linked H3K27 demethylase, is associated with stunted growth and CP, as observed in one patient with haploinsufficiency [140]. Zebrafish studies have confirmed that reduced *kdm6a* expression leads to craniofacial defects, supporting its role in CP formation [141]. Conditional knockout of *Kdm6a* in NCCs using *Wnt1*-Cre in mice revealed sex-specific effects, with females showing more severe phenotypes, including CP. Males may compensate for *Kdm6a* loss through a Y-linked homolog without demethylase activity. Despite the importance of Kdm6a in neural crest development, no changes in H3K27 or H3K4 trimethylation were observed, suggesting that *Kdm6a* regulates development through mechanisms independent of its demethylase activity [142].

*Kdm6b* is a critical player in cranial neural crest development, and loss of *Kdm6b* disrupts P53 pathway-mediated activity, resulting in a complete CP, along with cell proliferation and differentiation defects in mice. Kdm6b and Ezh2 antagonistically control H3K27me3 activity in the Trp53 promoter in cranial NCCs. More importantly, in the absence of *Kdm6b*, the transcription factor *Tfdp1*, which normally binds to the *Trp53* promoter, failed to activate the expression of *Trp53* in palatal mesenchymal cells. Furthermore, the expression of *Trp53* in such cells cannot be compensated for by the highly homologous histone demethylase Kdm6a [143].

##### Histone H3 Lysine 4 Methyltransferase KMT2D

Mutations in *KMT2D*, similar to those in *KDM6A*, are associated with Kabuki syndrome. A Xenopus model study showed that the knockdown of *kmt2d* affected NCC dispersion, but not other migratory behaviors, confirming the role of KMT2D in H3K4 methylation [144]. This study identified sema*3f*, a gene necessary for cranial NCC migration, as a target, and its overexpression partially rescued this phenotype. In contrast, a mouse conditional knockout (cKO) of *Kmt2d* showed a fully penetrant CP, but did not affect NCC migration. This observation is consistent with that observed in patients with Kabuki syndrome [145]. This cleft palate is associated with abnormal expression of extracellular matrix members. *Kmt2d* mutations may affect other cis genes, with downstream effects involving RAP1A dysfunction and impaired RAS/MAPK activation [146]. MAPK signaling inhibitors, such as desmethyl-dabrafenib, can prevent structural defects during embryogenesis in a zebrafish model of Kabuki syndrome, with any toxic effects [147], and may represent a future therapeutic strategy.

##### Histone-Lysine Demethylase PHF8

Mutations in histone demethylases such as PHF8, which demethylates H4K20 and H3K9, can lead to severe developmental disorders. PHF8 mutations are linked to CL/P and X-linked intellectual disabilities [148]. PHF8’s demethylase activity is essential for ribosomal RNA (rRNA) transcription [149] and neural differentiation [150]. Its catalytic domain, 2OG oxygenase, suggests a link between hypoxia and OFCs, particularly in children of mothers exposed to tobacco smoke during conception [151]. In zebrafish, *phf8* regulates *msx1* expression, which is associated with neural-crest induction and craniofacial development [150]. *Phf8* overexpression in mice also promotes bone regeneration, indicating potential therapeutic applications for craniofacial defects through regulation of special AT-rich sequence binding protein 2 *(SATB2)* [152].

##### Histone-Lysine N-Methyltransferase MECOM (PRDM3)

N-methyltransferase MECOM (PRDM3) regulates the methylation of H3K4 and H3K9 residues. During zebrafish development, *prdm3* is highly expressed in pharyngeal arches [153]. Morpholino knockdown results in neural and stellate defects and reduced expression of the NCC markers *dlx2a* and *barx1*. In another study, similar effects of *prdm3* knockdown were observed in developing zebrafish, with decreased methylation of H3K4 and H3K9 [154]. Conditional knockout of *Prdm3* by *Sox2*-Cre in developing mice results in mid-gestational lethality.

##### Histone-Lysine N-Methyltransferase PRDM16

Prdm16 has overlapping functions with *prdm3* in zebrafish, as its knockdown reduces the expression of neural crest markers *dlx2a* and *barx1*, and lowers H3K4 and H3K9 methylation [154]. Prdm16 is crucial for palatogenesis in mice, as shown by loss-of-function studies using mutagenesis, RNA interference, and gene trapping [155]. During palate development, *Prdm16* regulates several target genes involved in myogenesis, chondrogenesis, and osteogenesis [156], and its loss perturbs the expression of *Tgf-β* and *Bmp* signaling pathways. Conditional knockouts revealed its role in H3K9 methylation, but not H3K4 methylation [154], and it may also regulate orofacial development through *Smad* transcription factors [8].

##### Arginine Methyltransferase PRMT1

*PRMT1* encodes an arginine methyltransferase responsible for H4R3me2a modification, and regulates over 85% of arginine methylation activity, some of which targets non-histone proteins [157]. Conditional knockout of *Prmt1* in NCCs using *Wnt1*-Cre leads to craniofacial malformations, including CP, similar to defects observed in *Msx1-*null mice [158]. *Prmt1* cKO reduces *Msx1* expression in critical craniofacial regions at embryonic day 12.5. A follow-up study found disrupted BMP signaling in knockouts linked to *PRMT1* methylation of *Smad6*, a BMP inhibitor. Reduced H4R3me2a suggests *PRMT1*’s role in development also involves histone modification [159].

##### Histone Methyltransferase WHSC1

WHSC1 is a methyltransferase gene linked to WHS, and is expressed in both the epithelium and mesenchyme during mouse palate development. The expression was reduced when pregnant mice were treated with all-trans retinoic acid (ATRA), a treatment known to cause CP [160]. Researchers have suggested that *whsc1* plays a role in promoting cell proliferation. In a separate study, the knockdown of *whsc1* in Xenopus led to reduced facial width and smaller midfacial areas. It decreased the migratory distance and total area of the cranial NCCs [161].

##### Histone Deacetylases HDAC3 and HDAC4

HDAC3 is essential for mouse development, and conditional knockout in NCCs leads to craniofacial defects, including CP [162]. HDAC3 regulates the transcription factors *Msx1*, *Msx2*, and *Bmp4*. In conditional knockouts, these genes showed increased expression, whereas cell proliferation decreased and apoptosis increased at E12.5. Histone acetylation is likely important for balancing gene expression in NCCs.

HDAC4, a class II histone deacetylase, plays a role in osteogenesis by interacting with MEF2 and regulating endochondral ossification [8]. During zebrafish development, *hdac4* is expressed in pre-migratory and migrating cranial NCCs. *hdac4* knockdown results in reduced or absent cranial NCCs, leading to palatal defects such as shortened, clefted, or missing ethmoid plates [163].

##### Histone Acetyltransferase KAT6A

Microdeletion or mutation of *TBX1* in humans causes 22q11.2 microdeletion syndrome, which includes symptoms such as SMCP, heart defects, and thymic dysfunction [164]. In mice, the deletion of the acetyltransferase KAT6A, which regulates *Tbx1* expression, partially mimicked 22q11.2 microdeletion syndrome, including SMCP. The study found that an extra copy of *Tbx1* did not rescue the palatal defects caused by *Kat6a* deficiency, suggesting that either a higher dosage of *Tbx1* is needed or that *Kat6a* influences additional genes beyond *Tbx1*.

#### 3.4.5. Non-Coding RNAs in Craniofacial Development and Orofacial Clefts

miRNAs are small (~22-nucleotide) non-coding RNA molecules that regulate various developmental processes. miRNAs typically bind to the 3′ untranslated regions (3′UTRs) of target mRNAs through complementarity between the miRNA seed sequence and miRNA response element (MRE) [14]. Perfect binding results in mRNA degradation, whereas partial binding leads to suppression of transcription. miRNAs can regulate multiple mRNAs, and a single mRNA can be targeted by several miRNAs. In addition to gene silencing, miRNAs can activate transcription, upregulate protein expression, and target mitochondrial transcripts [14]. miRNAs play important roles in embryonic orofacial tissues by targeting genes involved in cell proliferation, apoptosis, differentiation, cell adhesion, and EMT (Table 3) [165].

The formation of craniofacial structures begins with migration and patterning of NCCs. Dicer, an enzyme essential for miRNA processing, is critical for the survival of post-migratory NCCs, although it is not required for their initial migration into facial primordia [166]. NCC-specific *Dicer* knockout mice exhibit various craniofacial defects, with some showing microcephaly, facial hypoplasia, and CP [166]. However, palatal development is halted in *Dicer* knockout (KO) models because NCC-derived skeletal structures remain immature or absent [166]. Most of these defects result from extensive apoptosis induced by NCC derivatives and are associated with defective MAPK/ERK signaling [166]. Functional analysis has revealed that *miR-21* and *miR-181a* repress *Sprouty2*, a negative regulator of the MAPK/ERK signaling pathway essential for cell proliferation, differentiation, and apoptosis [167]. Due to abnormal NCC patterning, murine *Dicer* disruption leads to severe defects in the maxillary, mandibular, and frontonasal processes. *miR-452* is key in regulating EMT and NCC patterning by targeting *Wnt5a* [168]. Loss of *miR-452* increases Wnt5a expression and reduces *Shh* and *Fgf8* signaling, thereby decreasing *Dlx2* expression, a key regulator of NCC patterning in the first pharyngeal arch [130]. Recently, *miR-149* has been implicated in the etiology of non-syndromic cleft lip with or without palate through its role in migrating human neural crest cells (hNCCs) derived from human induced pluripotent stem cells (Figure 6). Using 3′ RNA-Seq, 604 differentially expressed genes were identified in hNCCs overexpressing *miR-149* compared with untreated cells, highlighting their involvement in this process [169]. Ref. [170] demonstrated that CP model gene and miRNA expression from E10.5 to E14.5 in the maxillary processes to identify spatiotemporal patterns of gene and miRNA expression. *MiR-325-3p* and *miR-384-5p*, that repressed cleft-related genes *Adamts3*, *Runx2*, *Fgfr2*, *Acvr1*, and *Edn2*, while their expression increased over time. On the contrary, *miR-218-5p* and *miR-338-5p* repressed cleft-related genes *Pbx2*, *Ermp1*, *Snai1*, *Tbx2*, and *Bmi1*, while their expression decreased over time. The findings suggest that these miRNA mimics significantly inhibited the cell proliferation of the mouse palate mesenchymal cell and O9-1 cranial NCC line by modulating CP-associated genes, confirming their regulatory role in the pathogenesis [170].

During palatogenesis and OFC development, specific miRNAs play crucial roles in regulating cell proliferation (Figure 6). Overexpression of *miR-133b*, *miR-374a-5p*, and *miR-4680-3p* inhibits cell proliferation in human embryonic palate mesenchymal (HEPM) cells, probably by downregulating *GCH1*, *PAX7*, *FGFR2*, and *ERBB2* [14,171]. Similarly, *miR-497-5p* and *miR-655-3p* reduce the proliferation of human lip fibroblasts by targeting multiple genes implicated in OFCs in CL/P studies. *miR-124-3p* reduces myeloma cell line (MELM) cell proliferation by downregulating *Bmpr1a*, *Cdc42*, and *Tgfbr1* [172]. In particular, strong expression was found in the maxillary process of *miR-124-3p*, particularly at E13.5. In addition, the conserved human and mouse regulatory subnetwork with five transcription factors, including *GLI2*, *PAX3*, *PAX7*, *PAX9*, and *SATB2* [173]; three non-transcription factor genes, *FGFR1*, *RARA*, and *SUMO1*; and five miRNAs, including *miR-27b*, *miR-133b*, *miR-205*, *miR-376b*, and *miR-376c*, control cell proliferation of lip mesenchymal cells via the following specific gene targets: *miR-27b* targets *PAX9* and *RARA*; *miR-133b* targets *FGFR1*, *PAX7*, and *SUMO1*; and *miR-205* targets *PAX9* and *RARA* [174].

In addition, refs. [171,175] identified miRNAs (*miR-133b*, *miR-140-5p*, *miR-374a-5p*, *miR-381a-3p*, and *miR-4680-3p*) associated with CP development in humans through systematic reviews, bioinformatics analyses, and cell proliferation assays in human embryonic palatal mesenchymal (HEPM) cells [171,175]. Using data from patients with CP and HEPM cells, ref. [176] demonstrated that *hsa-let-7c-5p* and *hsa-miR-193a-3p* are involved in the development of CP using the data of CP patients and HEPM cells. Ref. [177] has shown that Genes predicted in all three databases (FUNRICH, MIRDB, and Targetscan) were considered as potential target genes of the miRNAs (104 target genes of *miR-193a-5p*, 512 target genes of *let-7c-5p*). In particular, PIGA and TGFB2 were selected as the most promising targets by further inquiring into three databases (MGI, MalaCards, and DECIPHER). Recently, ref. [178] found that Phenobarbital (PB) specifically induced *let-7c-5p* expression, and that the *let-7c-5p* specific inhibitor alleviated the PB-induced suppression of HEPM cell proliferation, indicating that *let-7c-5p* plays a crucial role in PB-induced toxicity. *Let-7c-*5p is highly expressed in craniofacial tissues of embryonic mice [178].

Studies of the *mir-17-92* cluster provided the first genetic evidence that specific miRNAs are functionally associated with mammalian CL/P [179]. The *mir-17-92* cluster, which consists of six highly conserved miRNAs (*miR-17*, *miR-18a*, *miR-19a*, *miR-19b-1*, *miR-20a*, and *miR-92a-1*) belonging to four families (*miR-17*, *miR-18*, *miR-19*, and *miR-92*), is located on mouse chromosome 14 (chromosome 13 in humans) [8]. The expression of *mir-17-92* and its two paralogs follows a similar pattern in mouse embryos, decreasing from E12 to E14 and concentrated in the distal tips of the PS during palatogenesis [165,179]. Direct targets of *miR-17-92* include T-box factors such as *Tbx1* and *Tbx3*, which harbor functional MREs in their transcripts. These genes are upregulated in *miR-17-92* mutant craniofacial structures, and studies have shown that *miR-17-92* directly represses *Fgf10* expression, which is crucial for proper maturation of the palate epithelium [8,179]. ChIP and ChIP-Seq data demonstrated the binding of *AP-2α* and *Smad1/2/5* to *miR-17-92* chromatin, suggesting that AP-2α and BMP signaling regulate *mir-17-92* expression. It is evident from these studies that the downregulation of specific progenitor genes, such as T-box factors, by miR-17-92, is critical for normal midfacial development [179].

The *miR-17-92* cluster also targets the TGF-β signaling pathway in the palatal mesenchyme [8]. A decrease in the expression of *miR-17-92* from E12–14, with a concomitant increase in the expression of key components of the pathway, such as *TGFBR2*, *SMAD2*, and *SMAD4*, was evident in the palatal shelves. Luciferase assays in palate mesenchymal cells (PMCs) revealed that *TGFBR2* is directly targeted by *miR-17* and *miR-20a*, whereas *SMAD2* and *SMAD4* are targets of *miR-18a* [8]. It is hypothesized that the *mir-17-92* cluster regulates palatal shelf elongation and elevation by regulating the TGF-β induced inhibition of proliferation and collagen synthesis. *E2F1* directly binds to the *miR-17-92* promoter, facilitating its transcription. The findings in palatal mesenchymal cells (PMCs) include the expression of *E2F1* and *E2F3* in palatal tissue on E12-14, significant inhibition of cell proliferation upon *E2F1* knockdown, and upregulation of the *miR-17-92* cluster with *E2F1* overexpression [180]. Excessive cell proliferation is curtailed when *miR-17* and *miR-20a* bind to the 3-UTR of *E2F1*, thereby forming a negative feedback loop that helps regulate the cell cycle. Abnormal regulation of this negative feedback loop may lead to palatal cleft formation [180].

Palatal fusion proceeds through the resolution of a medial edge seam (MES), which is consistent with a mechanism of “convergence and extrusion” during the final stages of palatal fusion [181]. It involves the creation of temporary epithelial bridges over the opposing palatal shelves, followed by a thick epithelial layer that converges into a unified monolayer through cellular intercalation and oronasal translocation of MES epithelium [181]. Ref. [182] has been identified *miR-22–3p*, an actomyosin contractility regulator during palatal fusion. Inhibition of *miR-22* activity in palate organ cultures using anti-*miR-22* resulted in the failure of MEE dissolution and MES removal, supporting a key role for *miR-22* in palatogenesis. Several potential mRNA targets of *miR-22* are transcripts encoding two myosin-heavy chains (*Myh9* and *Myh10*) and essential actomyosin contractility. Although functional interactions between *miR-22* and these targets have not been demonstrated in palatal tissues, mRNAs *Myh9* and *Myh10* represent functionally validated targets of *miR-22* in other biological settings [182].

The EMT transcription factor (TF) balance is important for efficient palatal closure. Balanced regulation of epithelial and mesenchymal TFs, such as *Grhl2* and *Zeb1*, is necessary for palatal closure [183]. *Zeb1* is considered responsible for mesenchymal identity, whereas Grhl2 and its targets (e.g., Ovol1, Ovol2, and the *miR-200* family of miRNAs) promote epithelial identity [183]. The results of this study indicated that *Grhl2/miR-200* and *Zeb1/Zeb2* antagonize each other, and that *Grhl2* transactivates the *miR-200* family of miRNAs to repress *Zeb1/Zeb2* [183]. In this respect, *miR-200b* was shown to target *Smad2*, *Snail*, *Zeb1*, and *Zeb2*, all of which encode transcription factors that function as mediators of the Tgf-β signaling pathway. In response to TGF-β, SMAD2/3 is activated. It forms a complex with SMAD4, which then interacts with either ZEB1, ZEB2, or SNAIL to repress epithelial markers, stimulate mesenchymal markers and induce migration and apoptosis [14,184]. *miR-200b* expression in MES gradually diminishes as palatal fusion proceeds, as the inhibition of cell proliferation in MES is necessary before palates can fuse. These results indicated that miR-200b overexpression induces abnormal palatogenesis in MES by inhibiting *Tgf-β*-mediated *Smad2* and *Snail* expression [14,184].

Analysis of the MEE of *Tgf-β3*^−/−^ mouse fetuses at E13.5, indicated that the expression of *miR-206* in the developing palate was significantly diminished compared to that in their wild-type counterparts, suggesting an important role of *miR-206* in palatal ontogeny [14,185]. Expression profiling of mouse embryonic maxillary mesenchymal (MEMM) cells treated with anti-*miR-206* revealed significant changes in the expression of ~230 genes, including several members of the TGF-β and Wnt/β-catenin superfamilies [185]. Aberrant regulation of this pathway can cause various disorders, such as OFCs, Kallman syndrome, Crouzon syndrome, and Apert syndrome [16].

Additionally, maternal miRNA expression influences cleft risk. For example, overexpression of *miR-152* has been associated with craniofacial dysmorphism, and maternal circulating *miR-let7-3p* may be a diagnostic biomarker for NSCLP (Table 3). SNPs in miRNA biogenesis enzymes or within miRNA-binding regions of cleft-related genes (e.g., FGF2, *FGF9*, and *MSX1*) have been associated with cleft risk [174]. A genetic association study showed that an SNP (rs7205289:C > A) located in the precursor of *miR-140* (*pre-mir-140*) contributes to non-syndromic CP susceptibility by influencing the processing of *miR-140* [177]. The minor A allele of rs7205289, with a higher frequency in patients, is associated with decreased *miR-140-5p* expression and increased *miR-140-3p* expression [175]. In addition, this allele is conserved in primates and functionally important. Therefore, dysregulation of *miR-140* may be implicated in the etiology of CP, which is supported by genetic evidence. miRNAs are essential for properly regulating craniofacial development, and this biomarker is supported by numerous craniofacial developmental defects, such as CP and other orofacial defects resulting from miRNA dysregulation [14].

Long non-coding RNAs (lncRNAs) act as miRNA sponges by binding to specific miRNAs via MREs to reduce miRNA levels [186]. The function of long non-coding RNAs (lncRNAs) is closely related to their subcellular localization. In the cytoplasm, lncRNAs play regulatory roles by functioning as miRNA sponges and engaging in a competing endogenous RNA (ceRNA) mechanism. Furthermore, cytoplasmic lncRNAs can interact with RNA-binding proteins (RBPs), thereby exerting biological effects [187]. In contrast, the nuclear localization of lncRNAs enables them to modulate gene transcription or pre-transcriptional processes through interactions with DNA promoter regions or transcription factors, referred to as cis- and trans-regulatory mechanisms (Table 3) [188,189].

**Table 3 ijms-26-01382-t003:** Regulatory roles of microRNAs (miRNAs) and long non-coding RNAs (lncRNAs) in orofacial development.

Molecule Type	Specific Molecule	Target/Pathway Affected	Effect on Orofacial Cleft Development	References
miRNA	*miR-21*, *miR-181a*	*Sprouty2*, (*MAPK/ERK* pathway)	cell proliferation, differentiation, and survival of neural crest cells	[167]
	*miR-452*	*Wnt5a*, *Shh*, *Fgf8* signaling, *Dlx2*	EMT and neural crest cells patterning	[130,168]
	*miR-149*	hNCC migration	neural crest cells	[169]
	*miR-325-3p*, *miR-384-5p*	*Adamts3*, *Runx2*, *Fgfr2*, *Acvr1*, *Edn2*	cell proliferation	[170]
	*miR-218-5p*, *miR-338-5p*	*Pbx2*, *Ermp1*, *Snai1*, *Tbx2*, *Bmi1*	cell proliferation	[170]
	*miR-133b*, *miR-374a-5p*, *miR-4680-3p*	*GCH1*, *PAX7*, *FGFR2*, *ERBB2*	cell proliferation	[14,171,174]
	*miR-497-5p*	*mTOR*	cell proliferation	[172]
	*miR-655-3p*	*TGF-β*, *Wnt*	cell proliferation	[172]
	*miR-124-3p*	*Bmpr1a*, *Cdc42*, *Tgfbr1*	proliferation in embryonic lip mesenchymal cells	[172,173]
	*miR-27b*	*PAX9*, *RARA*	cell proliferation of lip mesenchymal cells	[174]
	*miR-133b*	*FGFR1*, *PAX7*, *SUMO1*	cell proliferation of lip mesenchymal cells	[171,174,175]
	*miR-205*	*PAX9*, *RARA*	cell proliferation of lip mesenchymal cells	[174]
	*let-7c-5p*, *miR-193a-3p*	*PIGA*, *TGFB2*	cell proliferation	[174,176,178]
	*miR-17*, *miR-18a*, *miR-19a*, *miR-19b-1*, *miR-20a*, *miR-92a-1* (*mir-17-92* cluster)	*Tbx1*, *Tbx3*, *Fgf10*, *TGFBR2*, *SMAD2*, *SMAD4*	midfacial development	[8,165,179,180]
	*miR-17*, *miR-20a*	*E2F1*	MES	[180]
	*miR-22-3p*	*Myh9*, *Myh10*	MES dissolution and palatal fusion	[183]
	*miR-200b*	*Smad2*, *Snail*, *Zeb1*, *Zeb2 * (mediators of TGFβ signaling)	MES	[14,184]
	*miR-206*	*TGFβ*, *Wnt/β-catenin*	palatal fusion	[14,16,185]
	*miR-140*	SNP rs7205289, *TGF-β*	cell migration	[175,177]
	*miR-744-5p*	lncRNA RP11-462G12.2 (C-allele)	cell apoptosis, proliferation	[190]
lncRNA	RP11-462G12.2 (C-allele)	*miR-744-5p*, *IQSEC2*	C-allele binds *miR-744-5p*	[190]
	NONMMUT100923.1	*miR-200a-3p*, *Cdsn*	medial edge epithelial cell adhesion	[188,191]
	NONMMUT004850.2/NONMMUT024276.2	*miR-741-3p/miR-465b-5p*, *Prkar1α*	palatal fusion	[192]
	RP11-731F5.2, XIST, RP11-591C20.9	*miR-483-3p*, *miR-4690-3p*, *miR-654-3p*, *miR-6515-5p*, *RARA*, *SMPD1*	CL/P, CPO	[38,193]
	*MALAT1*	*miR-1224-3p*, *SP1*, *miR-6734-5p/miR-1224-3p*, *WNT10A*	non-syndromic orofacial clefts	[194]
	*NEAT1*	*miR-140-3p.1*, *CXCR4*, *miR-3129-5p/miR-199a-3p/miR-199b-3p*, *ZEB1*, *miR-130b-3p/miR-212-3p/miR-200b-3p*, *SMAD2*	non-syndromic orofacial clefts	[188]

The SNP rs2262251 (G > C) located in *lncRNA RP11-462G12.2*, was found to be specifically associated with CL/P but not CP. Overexpression of the G allele inhibits apoptosis and promotes the proliferation of HEK-293 and HEPM cells [14,190]. Transfection studies using luciferase reporters indicated that the C allele, but not the G allele, specifically binds to *miR-744-5p*. Transfection of *miR-744-5p* into HEPM cells decreased lncRNA expression of the C-allele, and this decrease was rescued using *miR-744-5p* inhibitors. Conversely, overexpression of lncRNA decreased *miR-744-5p* levels, thus confirming the sponging effect of lncRNA with the C-allele [190]. Both *miR-744-5p* and *IQSEC2* exhibited a significant reverse correlation and were expressed in lip tissues of patients with CL/P. Thus, the C-allele of the lncRNA regulates *IQSEC2* expression by sponging *miR-744-5p*, enhancing apoptosis, and suppressing proliferation [190]. *LncRNA-NONMMUT100923.1* was found to regulate *Cdsn* expression by competitively binding to *miR-200a-3p* in a ceRNA network during palatogenesis, potentially inhibiting medial edge epithelial cell adhesion by preventing desmosome junction disintegration [188]. For instance, one study highlighted the potential regulatory role of *NONMMUT004850.2/NONMMUT024276.2-miR-741-3p/miR-465b-5p-Prkar1α* in palatal fusion during CP development [192]. A complex regulatory association involving m*iR-483-3p*, *miR-4690-3p*, *miR-654-3p*, *miR-6515-5p*, *lncRNA RP11-731F5.2*, *lncRNA XIST*, *lncRNA RP11-591C20.9*, *RARA*, and *SMPD1* was also revealed in the CL/P and CPO groups [38,193].

Some studies utilizing the same lncRNA dataset GSE183527 as our study identified that ceRNA networks (MALAT1-hsa-miR-1224-3p-SP1, MALAT1-hsa-miR-6734-5p/hsa-miR-1224-3p-WNT10A, NEAT1-hsa-miR-140-3p.1-CXCR4, NEAT1-hsa-miR-3129-5p/hsa-miR-199a-3p/hsa-miR-199b-3p-ZEB1 [194], and NEAT1-hsa-miR-130b-3p/hsa-miR-212–3p/hsa-miR-200b-3p-SMAD2) may contribute to the etiology of NSOFCs [188]. lncRNA TPT1-AS1 has been reported to regulate various biological processes, including cell proliferation, apoptosis, autophagy, invasion, migration, and EMT, all of which are implicated in the progression of NSCL/P [188]. The potential roles of the lncRNAs FENDRR and TPT1-AS1 and the mRNAs EIF3H, RBBP6, and SRSF1 in NSCL/P development were investigated. Further investigation into the regulatory mechanisms of these lncRNAs and their interactions with other factors, such as environmental exposure, could provide additional insights into the development of CP [187,188].

#### 3.4.6. Epigenetic Regulation in Chromatin Organization and Craniofacial Development

The Polycomb repressive complexes PRC1 and PRC2 are essential transcriptional repressors that suppress gene expression [195]. EZH2, a catalytic subunit of *PRC2*, undergoes H3K27 methylation. Conditional knockout of *Ezh2* in murine NCCs causes Hox gene derepression, maintains NCCs in a pre-differentiation state, and impairs osteochondroprogenitor programs, which affect cartilage and bone formation [196]. *EZH2* mutations are also linked to Weaver Overgrowth syndrome, a genetic disorder that causes bone overgrowth [197]. Ring1b/Rnf2, an E3 ubiquitin ligase in PRC1, regulates craniofacial chondrocyte differentiation. Zebrafish ring1b mutants exhibit impaired cranial cartilage and bone development [198]. *PHC1* and *PHC2* are key regulators of *HOX* gene expression and craniofacial patterning, and genetic mutations affecting these proteins have been shown to underlie CATCH-22 syndrome associated with 22q11.2 microdeletion syndrome-like phenotypes and NSCLP [199]. ASXL proteins (ASXL1/2/3), scaffolding components of the PRC complex, are critical for early neural crest development. Mutations in ASXL cause neurodevelopmental disorders such as Bohring–Opitz, Shashi–Pena, and Bainbridge–Ropers syndromes, all of which feature craniofacial defects and developmental delays [200].

ATP-dependent chromatin remodeling complexes alter chromatin structure in an ATP-dependent manner to control gene expression, providing access to transcription factors and machinery associated with DNA. The SWI/SNF complex, which contains subunits including ARID1A, ARID1B, BRG1 (SMARCA4), BRM, BAF155, BAF170, SMARCA2 (SNF5), SMARCB1 (INI1), and SMARCE1, is an important regulator of gene expression. Mutations in these factors are associated with syndromic developmental defects, such as Coffin–Siris syndrome, which presents as a developmental disability, characteristic facial features, and malformations of the fifth finger or toe (Figure 5). Coffin–Siris syndrome patients frequently have mutations affecting the SWI/SNF complex subunits ARID1A or ARID1B. In mice, loss of Arid1a in NCCs causes severe craniofacial defects. In human studies, *ARID1B* mutations disrupt the switch between *ARID1A* and *ARID1B* during neural crest development, impairing differentiation [201]. *Brg1* mutations are also linked to Coffin–Siris syndrome, with loss of *Brg1* impairing NCC survival and differentiation. *Brg1* is essential for conserving a pool of multipotent NCCs, and its loss leads to craniofacial defects in zebrafish and mice [202]. Moreover, the loss of some core SWI/SNF subunits, such as BAF155 and BAF170, leads to abnormalities in the survival, migration, and differentiation of NCCs, thereby leading to craniofacial defects [203]. Other SWI/SNF complex subunits, including SMARCA2, SMARCB1, and SMARCE1, also play similar roles in neural crest specification and craniofacial development [204]. Another chromatin remodeler, CHD7, also plays a crucial role in neural crest development, and its mutations cause the CHARGE syndrome. CHD7 interacts with the PBAF complex to regulate the expression and migration of the neural crest. Animal models, including zebrafish and mice, show that mutations in Chd7 recapitulate craniofacial and cardiovascular defects found in CHARGE syndrome [30].

Other epigenetic regulators, such as SATB2, are also important in craniofacial development besides ATP-dependent chromatin remodeling complexes. *SATB2* is a chromatin reader that binds to the nuclear matrix attachment regions to activate transcription. In humans, translocations involving the 2q32–q33 chromosomal region, which encompasses *SATB2*, are associated with CP [106]. Mutations in *SATB2* are linked to *SATB2*-associated syndromes characterized by intellectual disability, developmental delays, and craniofacial and dental abnormalities [205]. In mice, *SATB2* mutations result in craniofacial abnormalities resembling human phenotypes and osteoblast differentiation defects [106]. *SATB2* regulates skeletal patterning by controlling *Hox* genes and later drives osteogenesis by regulating genes such as *Runx2* and *ATF4*, as well as craniofacial patterning genes including *Pax9*, *Msx1*, *Alx4*, and *Lhx7*. These findings show that *SATB2* plays a dual role in neural crest development: it promotes NCC survival early on and later drives osteogenic differentiation during craniofacial structure formation.

#### 3.4.7. Environmental Influences on Epigenetics and Craniofacial Development

Exposure to environmental factors, especially those that take place in utero, is associated with a higher risk. These include smoking tobacco, drinking alcohol, and/or using several drugs such as benzodiazepines, corticosteroids, antibiotics, and antiepileptic drugs [206]. Additionally, organic solvents and pesticides have been implicated in occupational exposure [207]. It has been postulated that the interaction between genetic predispositions and environmental factors leads to CP development (Figure 6) [206,208]. A recent study demonstrated an increased risk of OFCs due to the high concentrations of toxic elements in diverse biological matrices. However, contiguous relationships were found for elevating the concentrations of essential trace elements (ETEs) to reduce the risk. Dietary intake of maternal foods containing ETEs, such as zinc (Zn), selenium (Se), copper (Cu), cobalt (Co), and molybdenum (Mo), is associated with a marked decrease in the risk of OFCs [207]. These results imply that exposure to toxic elements, both environmental contaminants and food, contributes to the risk of OFCs. In contrast, ETEs are inversely associated with OFCs, indicating a protective role against OFCs [207].

RA, a metabolic derivative produced through the retinol (vitamin A) metabolic pathway, is an important signaling molecule required for both morphogenesis and differentiation during embryonic development, including craniofacial morphogenesis. Deficiencies in vitamin A and its metabolites have long been known to cause congenital craniofacial defects, and excess retinoids induce defects in rodents, including CL/P [16,209]. In rodents, studies on the role of RA signaling have utilized various approaches, such as targeted knockout of retinoic acid receptors (*RARs*) and retinoid X receptors (*RXRs*), knockout or inhibition of enzymes involved in RA metabolism, and addition of exogenous RA [206]. Circulating retinol is bound by a retinol-binding protein (RBP) and taken up by target cells via the RBP receptor, STRA6 [210]. Cellular retinol-binding protein (CRBP) facilitates the conversion of retinol to retinaldehyde by alcohol dehydrogenase (ADH) or retinol dehydrogenase (RDH) enzymes, which are then oxidized to RA by retinaldehyde dehydrogenases (RALDH1-3) [211]. RA activates *RARs* and *RXRs*, which regulate gene transcription [206]. Mutations in *RARs* (α, γ) result in CP and midfacial clefts, whereas *Aldh1a3*-deficient mice exhibit reduced RA activity leading to lethal choanal atresia [212]. All three *Cyp26* genes (*Cyp26a1*, *Cyp26b1*, and *Cyp26c1*) that metabolize RA are dysregulated in *Tbx1* mutants [213,214], leading to RA accumulation and reduced expression of the key regulators *Bmp2* and *Fgf10*, resulting in CP [215]. RA negatively modulates *Tgf-β* signal transduction by enhancing *Smad7* expression and downregulating *Smad2* activity, whereas *Tgf-β3* represses RA signaling. Two pathways mutually regulate the common co-repressors of TGIF1, RA, and TGF-β signaling [206,216]. ATRA, a form of RA, also mediates its effects through Notch signaling in palatal cells to drive apoptosis and cell proliferation. In previous studies, ATRA exposure increased *Notch1* expression in epithelial cells and *Notch2* in mesenchymal cells while reducing CyclinD1, inhibiting proliferation [217]. It also decreases the expression of key genes (*Lhx8*, *Msx1*, and *Msx2*) in chick embryos, affecting orofacial development [218]. Mutual repression of RA and Wnt signaling is crucial for craniofacial development [219]. *Wnt*-deficient mice show increased RA activity, whereas RA suppresses *Wnt/β*-catenin signaling and disrupts cell cycle regulation [206]. RA also maintains *Shh* and *Fgf8* expression, but excess RA inhibits *Shh* signaling and increases apoptosis in mouse maxillary tissue. *Shh* regulates RA levels through *Cyp26* gene activation, highlighting the complex interactions between the RA, Shh, and Wnt pathways in craniofacial development [209,219]. Notch signaling in both the epithelium and mesenchyme may be altered after ATRA exposure, thus altering p21 signaling. For example, ATRA exposure at E10 increased *Notch2* and decreased *CyclinD1* in mesenchymal cells between E12.5-14.5, inhibiting proliferation [217]. Another study found that ATRA exposure at E12 increased *Notch1* expression in MEE cells at E15, concomitantly inhibiting normal epithelial apoptosis [220]. ATRA also reduced the expression of *Lhx8*, *Msx1*, and *Msx2* in the upper jaw of chick embryos and genes regulated by *Fgf* and RA signaling [218]. ATRA may influence Lef1-mediated periderm EMT in mice by decreasing *Ambra1* expression, whereas *Cyp26b1* is essential for proper palatogenesis and tongue depression [221]. Ref. [222] has showed that ATRA administration induced changes in miRNA in mice with CP. The CP in RA-treated mice has been linked to the upregulation of *miR-181a-5p*, *miR-410-3p*, *miR-3960*, *miR-1224-5p*, *miR-3970*, *let-7e-5p*, *miR-1907*, *miR-124–3p*, and *miR-4680–3p* [185,222]. Then, ref. [222] demonstrated that *miR-140-3p*, *miR-351-5p*, and *miR-503-5p* were downregulated in a mouse model of RA-treated (Figure 6). In ATRA-treated mouse embryonic palatal mesenchymal (MEPM) cells and embryos, *miR-124–3p* levels are increased, leading to the downregulation of target genes such as *Axin1*, *Fst*, *Vcan*, and *Zeb1* [175]. Also, the previous study showed that *miR-129-5p* and *miR-340-5p* repress cell growth of primary mouse embryonic palatal mesenchymal cells and neural crest-derived O9-1 cells. Specifically, *miR-129-5p* targets *Sox5* and *Trp53*, while *miR-340-5p* regulates the expression of *Chd7*, *Fign*, and *Tgfbr1* to modulate cell growth [223]. *MiR-106a-5p* was significantly (~8.9-fold) upregulated in ATRA-induced CP tissues and the overexpression of *miR-106a-5p* downregulated the expression of *Tgfbr2* and altered the levels of *pSMAD2* and *pSMAD3* [224]. In addition, *miR-106a-5p* was strongly negatively correlated with cholesterol metabolites, suggesting that *miR-106a-5p* may play a putative role in cholesterol metabolism through TGF-β signaling in the defective palatogenesis of ATRA-induced CP [224]. Thus, the ability of *miR-106a-5p* to modulate apoptosis may occur via the TGFβ/SMAD pathway [14,224]. Recently, a ceRNA regulatory network was elucidated, where *LncRNA-NONMMUT100923.1* regulates *Cdsn* expression by competitively binding to endogenous *miR-200a-3p* during palatogenesis in an ATRA induced murine model [191]. Recently, RNA sequencing of mouse embryonic palatal shelf (MEPS) tissue revealed that ATRA treatment significantly upregulated *miR-470-5p* expression while downregulating Fgfr1 expression compared to controls. The results consistently demonstrated that *miR-470-5p* inhibits EMT in MEPS epithelial cells, primarily by suppressing *Fgfr1* expression [225].

Alcohol drinking has been reported to be associated with an increased risk of OFC in studies from China, the Democratic Republic of the Congo, and Mexico [206,226]. Notably, a pooled study indicated an increased risk of CLO associated with binge drinking during pregnancy [206,227]. The exact mechanisms by which alcohol increases the risk of OFCs are not fully understood; however, several teratological hypotheses have been proposed. These include antagonism of RA signaling, altered epigenetics, and oxidative stress, with ethanol (EtOH) oxidized to the toxic intermediate acetaldehyde (AcAL) [228]. These mechanisms involve the oxidation of EtOH into a toxic intermediate, AcAL, which is ultimately metabolized to acetyl-CoA by aldehyde dehydrogenase (ALDH) [228].

In the RA-EtOH competition model, AcAL competes with retinaldehyde for the enzyme synthesizing RA. Research using Xenopus has shown that EtOH exposure inhibits RA synthesis by preventing the activity of Aldh1a2, which converts retinaldehyde into RA [229]. Kinetic analysis of human ALDH1A2 revealed a preference for AcAL over retinaldehyde as a substrate, providing a biochemical basis for the RA-EtOH competition model. Another zebrafish study found that craniofacial malformations in *pdgfra* mutants were exacerbated by EtOH and that pdgfra protected against EtOH through the PI3K-mTOR pathway [230]. EtOH can alter the epigenetic state of neural stem cells, affecting the methylation of cell cycle genes and exacerbating malformations in genetic mutants [231]. Additionally, EtOH metabolism to AcAL by CYP2E1 can lead to oxidative stress, apoptosis, and activation of signaling pathways that could disrupt orofacial development [232]. In murine embryonic stem cells, EtOH and AcAL promote differentiation through transcriptional activities of the RA receptor, RAR-γ, which induces the expression of target genes such as Hoxa1 and Cyp26a1 [233]. Thus, EtOH exposure may alter cell stemness and/or fate during embryogenesis [234]. Although these effects are seemingly consistent with the disruption of orofacial development, there is little experimental evidence linking them. Much work remains to be performed regarding EtOH exposure and the mechanisms of OFC, but there is sufficient knowledge to begin well-planned investigations [206].

Dioxins and related compounds are a group of chemically similar substances consisting of two coplanar benzene rings, known to induce various toxicity phenotypes with differing potencies [14]. Each compound was assigned a toxic equivalency factor (TEF) based on its toxicity relative to TCDD, which is the most toxic compound in the group. TCDD serves as a prototypical compound in studies of toxicity mechanisms, with teratogenicity being a sensitive indicator of its effects in experimental animals. TCDD exposure disrupts gene expression in both epithelial and mesenchymal tissues, affecting multiple stages of palatogenesis, including growth and fusion [206]. Experimental studies have shown that TCDD administration to pregnant hamsters, mice, and rats induces CP in their fetuses [206]. The aryl hydrocarbon receptor (AhR), also known as dioxin receptor, is a crucial ligand-activated transcription factor for the teratogenic effects of TCDD [235]. AhR is highly active in the epithelial cells of the palatal shelves, and is found in the bone and muscle tissues of the palate. In TCDD-exposed fetuses, the MEE is covered by a thinner epithelial monolayer with fewer filopodia, impairing proper fusion of the palatal shelves, unlike control fetuses, which have thicker epithelial layers [236]. Decreases in filopodia and fusion failure have also been observed in ex vivo cultures of palatal shelves exposed to TCDD [237]. Notably, TGF-β3 supplementation rescued the fusion of these cultures, supporting the implication of its insufficiency in TCDD-induced CP. However, the expression of TGF-β3 is increased in the palates of TCDD-exposed mice, highlighting the need for different experimental systems [238]. Additionally, TCDD exposure reduces the expression of several crucial factors involved in craniofacial development, including FGFR1, Runx2, osteopontin (OPN), MyoD, and desmin. It alters the expression of E-cadherin and Sox9, which may further contribute to CP formation [14,206]. Moreover, TCDD disrupts growth factor signaling during palatogenesis via AhR signaling, particularly by reducing the expression of TGF-α in human palatal epithelial cells [239]. This suggests that TCDD targets the epidermal growth factor (EGF) pathway, a conclusion supported by knockout studies, despite the seemingly counterintuitive decrease in ligand expression [206]. In human fetal palatal epithelial cells (hFPECs), TCDD exposure stimulates EGF receptor phosphorylation, leading to ERK/p38 phosphorylation and increased cell proliferation [240]. TCDD also promotes cell cycle progression to S and G2/M phases via the PI3K/AKT pathway [240]. Notably, exposure to TCDD disrupts the epithelial-to-mesenchymal transition by decreasing epithelial markers, such as E-cadherin and keratin-14, and increasing the expression of mesenchymal markers, such as vimentin and fibronectin. This is mediated by the induction of Slug, an inducer of EMT and a repressor of E-cadherin, which contains DREs in its promoter, modulated by AHR. This suggests that inappropriate EMT through Slug is a key mechanism in AHR-mediated CP [206,240]. This aligns with previous findings that TGF-β3 protects against CP following TCDD exposure [241]. Ref. [242] revealed that TCDD function is mainly related to the metabolic processes of intracellular compounds, including the metabolic processes of cellular aromatic compounds and the metabolism of exogenous drugs by cytochrome P450. Furthermore, circRNA_1781/miR-30c-1-3p/PKIB and XR_380026.2/miR-1249-3p/DNAH10 ceRNA networks were hypothesized to be involved in palatal development, suggesting that circRNA_1781/miR-30c-1-3p/PKIB and XR_380026.2/miR-1249-3p/DNAH10 ceRNA networks may be critical for palatogenesis, providing a foundation for the investigation of CP [242]. Recently, the upregulation of miRNAs (*miR-214-3p*, *miR-296-5p*, and *miR-33-5p*) in the TCDD-treated group, while miRNAs (*miR-92a-3p*, *miR-126a-3p*, and *miR-411-5p*) were significantly downregulated. Notably, qRT-PCR testing confirmed a significant difference in *miR-214-3P* expression. These findings support that TCDD inhibits palatal mesenchymal cell proliferation and migration through *miR-214-3p*, suggesting a its role in TCDD-induced CP in mice [243].

Dexamethasone (DEX) is a synthetic glucocorticoid (GC) used clinically in many applications owing to its anti-inflammatory and immunosuppressive properties. Such effects are mediated through interactions with various signaling pathways and molecules, including toll-like receptors and mitogen-activated protein kinases (Figure 6) [244]. The mechanism of action of GCs involves the diffusion of extracellular GCs into the cytoplasm, where they bind to the cytosolic GC receptor (GR). Without GCs, GR forms a complex with heat shock proteins (HSP70 and HSP90), FKBP52, and p23. The binding of GCs results in the dissociation of this complex, and the resultant GC-GR complex forms a dimer. The activated dimer then translocates to the nucleus and binds to glucocorticoid response elements (GREs) in the promoter region of target genes, thus activating transcription (transactivation) [244]. Alternatively, the active GC-GR complex can directly bind as a monomer to NF-κB (p50/p65), without dimerization. This monomeric complex, bound to NF-κB, subsequently binds to the NF-κB response elements to repress transcription. Despite their therapeutic benefits, GCs exhibit teratogenic and toxic effects. For example, the risk of CL/P, preterm birth, and low birth weight is increased two- to nine-fold after exposure to oral or systemic corticosteroids during pregnancy [174]. DEX can cross the blood–placental barrier and bind to cytoplasmic GR, inducing CP in mice by inhibiting cell proliferation in the palatal mesenchyme [244]. Recently, analysis of miRNA expression in developing mouse palatal shelves revealed distinct changes between embryonic days E13.5 and E14.5. The *miR-449* family (including *miR-449a-3p*, *miR-449a-5p*, *miR-449b*, *miR-449c-3p*, *and miR-449c-5p*) showed increased expression on E4.5, whereas *miR-19a-3p*, *miR-130a-3p*, *miR-301a-3p*, and *miR-486b-5p* showed decreased expression. The functional role of these miRNAs in cell proliferation was further investigated, demonstrating that overexpression of the *miR-449* family and *miR-486b-5p* represses cell proliferation in primary mouse embryonic palatal mesenchymal cells and the *O9-1* cranial NCC line. In contrast, when *miR-130a-3p* and *miR-301a-3p* were inhibited, the opposite effect was observed; these miRNAs’ lower expression also reduced cell proliferation in the same cell types [245]. These findings suggest that *miR-130a-3p* plays an important role in dexamethasone-induced CP in mice [174,245].

Folate metabolism is an important step during embryogenesis. Folate metabolism involves complex mechanisms of regulation and interaction with the methionine cycle. Food folic acid is subjected to metabolic processing in the small intestine and liver by first reducing it to dihydrofolate and then to tetrahydrofolate (THF) by dihydrofolate reductase (DHFR). Under the action of serine hydroxymethyltransferase (SHMT), vitamin B6 is required to produce 5,10-methylene-THF, followed by 5-methyl-THF by methylenetetrahydrofolate reductase (MTHFR). 5-Methyl-THF enters target cells via folate carriers and is converted to THF by methionine synthetase, using vitamin B12 as a cofactor. In this reaction, Hcy is converted into methionine. Methionine is then converted to S-adenosylmethionine, the major methyl donor in biomolecule methylation, which includes DNA and histone modifications. After donating a methyl group, SAM becomes SAH, which is recycled to homocysteine and completes the methionine cycle [206]. A systematic review showed an inverse association between folic acid supplementation and OFCs [206]. A cross-sectional study among the California population found a decreased prevalence of both CL/P (PR = 0.91, 95% CI: 0.82, 1.00) and CPO (PR = 0.81, 95% CI: 0.70, 0.93) following mandatory grain fortification with folic acid [246]. A recent meta-analysis found that folic acid supplementation was only protective when administered periconceptionally (OR = 0.64, 95% CI: 0.56, 0.74) versus during pregnancy (OR = 0.90, 95% CI: 0.71, 1.14) [247]. Recent studies have highlighted the gene-environment (G × E) interactions affecting the Chilean population’s OFCs and folate metabolism. A polymorphism in *SHTM1*, less frequent in CL/P cases, may protect against CL/P by reducing enzymatic activity and increasing cellular folate levels [248]. Folate is critical for DNA methylation because the folate cycle provides methyl groups to SAM, a molecule known to methylate DNA and histones via DNMTs and histone methyltransferases, respectively [112]. Similarly, three intronic *MTR* alleles involved in SAM metabolism might be protective [248]. The usual MTHFR c.677C > T polymorphism decreases the enzymatic activity of MTHFR and increases CL/P risk, particularly when combined with low maternal folic acid intake [249]. This result was corroborated by a meta-analysis that pooled 15 studies; maternal 677C > T mutations confer susceptibility to CL/P, confirming that maternal folate metabolism plays an important role in the development of OFCs [249]. Some theories have focused on folate facilitating correct DNA and histone methylation during development [244], whereas others believe it aids cell proliferation [250]. Study [251] tested the hypothesis that the prevention by folic acid is an epigenetic modification, specifically by determining whether changes in DNA methylation are associated with CL/P. Using the Illumina^®^ Human Beadchip 450K array (https://bioconductor.org/packages/devel/data/annotation/html/IlluminaHumanMethylation450kanno.ilmn12.hg19.html, accessed on 10 January 2025), Genomic DNA methylation levels were assessed from archived newborn bloodspots in a 182-member case–control study. CL/P cases demonstrated significant epigenome-wide hypomethylation compared to controls; 63% of CpG sites showing reduced methylation in case newborns. This was consistent across racially stratified subgroups. A total of 28 CpG sites reached epigenome-wide significance, all showing hypomethylation in cases. The most significant CL/P-associated differentially methylated region included the *VTRNA2-1* gene, which was also hypomethylated in cases (FWER *p* = 0.014) [251]. This region has been previously described as a nutritionally responsive, metastable epiallele. CL/P-associated methylation changes were generally more pronounced at or near putative metastable epiallelic regions. Gene Set Enrichment Analysis of CL/P-associated DMRs revealed an over-representation of genes involved in palate development, such as WNT9B, MIR140, and LHX8. These DNA methylation changes may partly explain the mechanism by which OFCs are responsive to maternal folate levels [251]. Polymorphisms in other folate-related genes have also been linked to OFCs. Folate deficiency impairs DNA repair owing to uracil misincorporation and contributes to genomic instability [206]. Folate also affects processes dependent on S-adenosylmethionine (SAM), including the synthesis of polyamines for cell proliferation, differentiation, apoptosis, and DNA methylation with respect to epigenetic regulation [206]. Folate deficiency can result in toxic homocysteine accumulation, with hyperhomocysteinemia associated with OFCs. Elevated homocysteine levels increase asymmetric dimethylarginine, reactive oxygen species (ROS), and oxidative stress [252]. Thus, folate exerts its protective effects by its involvement in cell growth and differentiation and in maintaining physiological balance.

Phenytoin is also known to be a teratogen. Between 1993 and 2007, approximately 1% of the population received antiepileptic drug prescriptions, and approximately 20% of these prescriptions were for phenytoin (Mayne Pharma Pty Ltd, Mulgrave, Australia). This percentage increases to approximately 5% in women of childbearing age, emphasizing this well-recognized teratogen’s importance. Women with epilepsy who take phenytoin either alone or in combination with other medications have a two- to threefold increased risk of having children with congenital malformations [253]. The fetal hydantoin syndrome, associated with phenytoin exposure, includes growth restriction, typical facial features that include midfacial hypoplasia, increased risk for cleft lip, limb abnormalities that most commonly consist of hypoplasia of the distal phalanges with small nails, and an increased incidence of heart defects. First-trimester exposure to phenytoin increases the risk of maxillary hypoplasia. In one study, the prevalence was 16.7% [254]. The relationship between maxillary hypoplasia and CL is considered an interrelated abnormality arising from the underdevelopment of the maxillary process. Phenytoin at teratogenic doses reduces the rate of the early embryonic heart of pregnant rats and causes an increase in the length of time embryonic hypoxia is sustained, which would explain the observed findings that hypoxia elevates and hyperoxia lowers the incidence of CL in sensitive mouse strains. The mechanism underlying this effect has been suggested to be the phenytoin’s inhibition of the HERG potassium channel along with its reduction of sodium and calcium channels, which could also explain the action on the embryonic heart [255]. Investigations in rat models of fetal hydantoin syndrome have consistently observed both cleft lip and maxillary hypoplasia within the same litter, suggesting a continuum in severity [254]. In addition, ref. [255] found that phenytoin, an inducer of CL, decreased cell proliferation through *miR-196a-5p* induction. During E10.5 to E12.5, *miR-196a-*5p expression significantly declines in the MxPs and NPs, while phenytoin upregulates *miR-196a-5p* and suppresses MELM cell proliferation by downregulating *Pbx1*, *Pbx3*, and *Rpgrip1l*. Ref. [256]. Especially, treatment with a specific inhibitor for *miR-196a-5p* restored cell proliferation through normalization of expression of CL-associated genes in the cells treated with phenytoin [255]. *miR-196a-5p* suppresses cell proliferation and promotes osteogenic differentiation in human Wharton’s jelly umbilical cord stem cells (WJCMSC) and suppresses bone formation in WJCMSC-sheet transplanted rat calvaria through suppression of *Serpinb2* [256]. These collective findings suggest a pleiotropic role for *miR-196a-5p* in diverse developmental processes, including those critical for palate formation. Recently, ref. [257] has shown that phenytoin inhibits HEPM cell proliferation in a dose-dependent manner by downregulating cyclin-D1 and cyclin-E phenytoin treatment upregulated *miR-4680-3p* while downregulating its target genes, *ERBB2* and *JADE1*. These findings indicate that phenytoin suppresses cell proliferation by modulating *miR-4680-3p* [257]. 

Maternal smoking was recognized as a risk factor in a meta-analysis of 24 case–control and cohort studies published in the World Health Organization Bulletin in 2004, reporting associations between maternal smoking and either CL/P [RR = 1.34; 95% CI: 1.25, 1.44] or CPO (RR = 1.22; 95% CI: 1.10, 1.35) [258]. Ten years later, the 50th Anniversary United States Surgeon General’s report declared sufficient evidence for a causal link between maternal active smoking and OFC [259]. Humans possess two *NAT* genes, *NAT1* and *NAT2*, each with over two dozen known polymorphisms that may affect arylamine detoxification and OFC risk. A study on a California population linked two fetal polymorphisms of *NAT1* that increased the risk of CL/P fourfold when mothers smoked during early pregnancy (compared to reference genotype infants with nonsmoking mothers [260]). Smoke is known to contain endogenous tobacco plant compounds such as nicotine, pyrolysis products, and added chemicals. Correspondingly, there are several hypothesized mechanisms by which tobacco smoke increases the risk of OFCs. Genetic susceptibility significantly influences the risk of tobacco smoke-induced OFCs. Polymorphisms in genes encoding developmental signaling ligands, such as TGF-α, TGFβ3, and BMP4, have been linked to OFCs in relation to maternal smoke exposure [206]. Variants affecting the interaction between TGF-α and smoking (gene-environment interactions, GxE) have been observed in several populations [206,261]. TGF-B3 encodes a ligand crucial for TGF-β signaling, essential for lip and palate development. Polymorphisms in TGFβ3 associated with smoking have been linked to CL/P, CPO, and SMCP [262]. BMP4 is a ligand for BMP signaling, has polymorphisms associated with CL/P, and plays a role in the fusion of medial and lateral nasal processes. Variants of BMP4 in conjunction with smoking have also been associated with CL/P [263]. The gene expression in cell cycle regulation, DNA repair, and oxidative stress response is affected by tobacco smoke in fetal mouse tissues [264]. Additionally, tobacco smoke can induce proteasome-mediated degradation of proteins that mediate DNA methylation in first branchial arch (BA1) cells [265]. Nicotine is a vasoconstrictor that can impair uterine vascular function and affect the blood flow and oxygen delivery to the fetus [266]. Several teratogens, such as polycyclic aromatic hydrocarbons (PAHs), dioxins, carbon monoxide, pesticides, and heavy metals such as cadmium, may be present in tobacco smoke [267]. Exposure to heavy metals can cause OFCs in rodent models and is believed to act teratogenically through the induction of oxidative stress and perturbation of redox-sensitive signaling pathways [268].

## 4. Conclusions

This comprehensive review illustrates the complex interaction of genetic, epigenetic, and environmental risk factors in regulating palatogenesis and etiology of CL/P. These studies have also elucidated the complex molecular networks associated with critical signaling pathways, including TGF-β, BMP, FGF, Wnt, and SHH, as well as epigenetic mechanisms (e.g., non-coding RNAs, miRNAs, and long non-coding RNAs) that contribute to craniofacial development. Recent evidence has elucidated the necessity for the precise regulation of various developmental stages, including proliferation, migration, differentiation, and apoptosis.

An additional layer of complexity arises from environmental factors, including maternal tobacco use, alcohol exposure, folate deficiency, and teratogens, such as RA and TCDD. This disruption leads to an imbalance between genetic and environmental factors that initiate normal development, resulting in various congenital malformations, including CL/P. Recent advances in gene-environment interactions and epigenetic regulation will illuminate future prevention and treatment methodologies. The remaining gaps need to be resolved in future studies. In particular, the role of non-coding RNAs and chromatin remodeling in craniofacial development awaits further investigation. Additionally, it is important to identify precise molecular targets to translate these findings into clinical applications. This review integrates the genetic, epigenetic, and environmental perspectives to establish a framework for future multidisciplinary strategies to reduce the burden of OFCs.

## Figures and Tables

**Figure 1 ijms-26-01382-f001:**
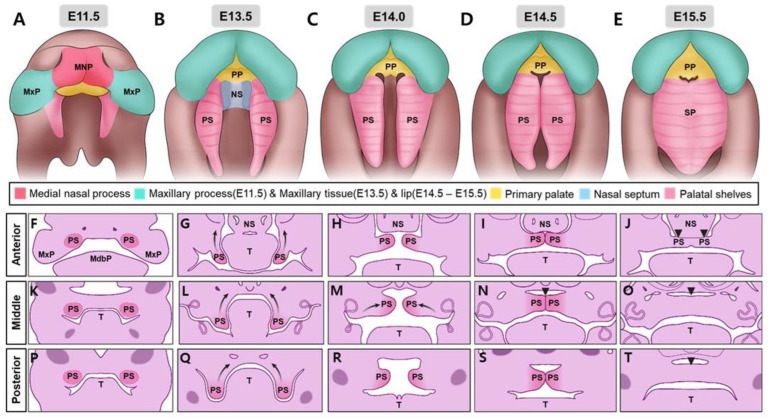
Developmental progression of secondary palate formation in mouse embryos from E11.5 to E15.5. (**A**–**E**) Frontal views show the developmental sequence of palatal shelf elevation and fusion. At E11.5 (**A**), the medial nasal process (MNP) and maxillary processes (MxP) are visible with the initial formation of the primary palate (PP). From E13.5 to E15.5 (**B**–**E**), the palatal shelves (PS) undergo vertical-to-horizontal elevation, with concurrent development of the nasal septum (NS). Progressive fusion of the PS occurs, resulting in the formation of the secondary palate (SP) by E15.5. (**F**–**T**) Frontal sections through the developing palate’s anterior, middle, and posterior regions at corresponding developmental stages. The sections demonstrate the progressive growth, elevation, and fusion of the PS around the tongue (T). The medial edge epithelium (MEE) is visible in anterior sections at early stages. The dynamic process of palatal shelf reorientation and fusion proceeds in an anterior-to-posterior sequence, with complete fusion achieved by E15.5. Color code: pink—medial nasal process (MNP); turquoise—maxillary process (MxP) (E11.5) and maxillary tissue (E13.5–E15.5) and lip; yellow—primary palate (PP); light blue—nasal septum (NS); light pink—palatal shelves (PS); black arrowheads indicate the gap between the primary and secondary palates, which will close following fusion between these tissues. Also, the black curved arrows indicate the direction of palatal growth.

**Figure 2 ijms-26-01382-f002:**
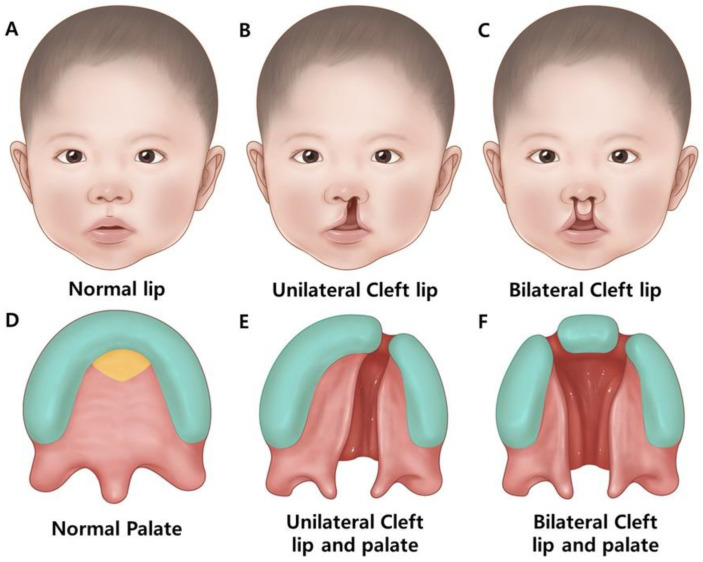
The anatomical spectrum of normal and cleft lip and/or palate malformations in humans. (**A**–**C**) Frontal facial views show variations in lip formation. (**A**) Normal lip morphology with complete fusion. (**B**) The unilateral cleft lip is showing incomplete fusion on one side. (**C**) Bilateral cleft lip presenting incomplete fusion on both sides of the upper lip. (**D**–**F**) Oral views of the palatal region depicting normal and cleft phenotypes. (**D**) Normal palate showing complete fusion of palatal shelves. (**E**) Unilateral cleft lip and palate with the incomplete fusion of the palatal shelf on one side. (**F**) Bilateral cleft lip and palate showing incomplete fusion of palatal shelves on both sides. Color code: turquoise—maxillary tissue; yellow—primary palate; pink—palatal and soft tissues.

**Figure 3 ijms-26-01382-f003:**
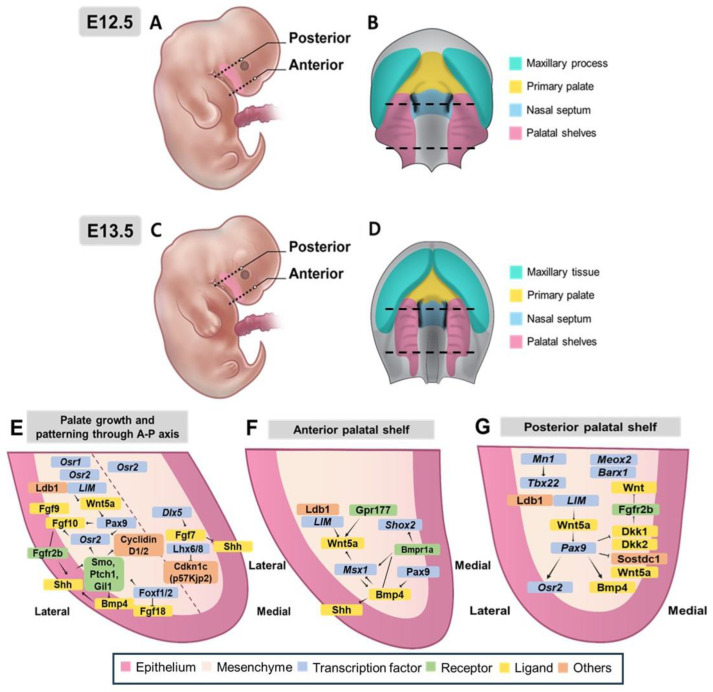
Complex molecular networks that coordinate palatal shelf growth, patterning, and morphogenesis along both the anterior–posterior and medial–lateral axes during palatogenesis. (**A**–**D**) Lateral and oral views of developing mouse embryos at E12.5 and E13.5 with corresponding schematic diagram The anterior and posterior orientation is indicated by dashed lines. (**E**–**G**) Schematic representations of molecular networks controlling palate development: (**E**) Key factors involved in palate growth and patterning along the anterior–posterior axis, showing interactions between epithelial and mesenchymal factors and their downstream targets. (**F**) Molecular regulation of anterior palatal shelf development. (**G**) Posterior palatal shelf patterning network showing interactions. Color code: pink—palatal epithelium; apricot—mesenchyme; blue—transcription factors; green—receptors; yellow—ligands; orange—other regulatory molecules.

**Figure 4 ijms-26-01382-f004:**
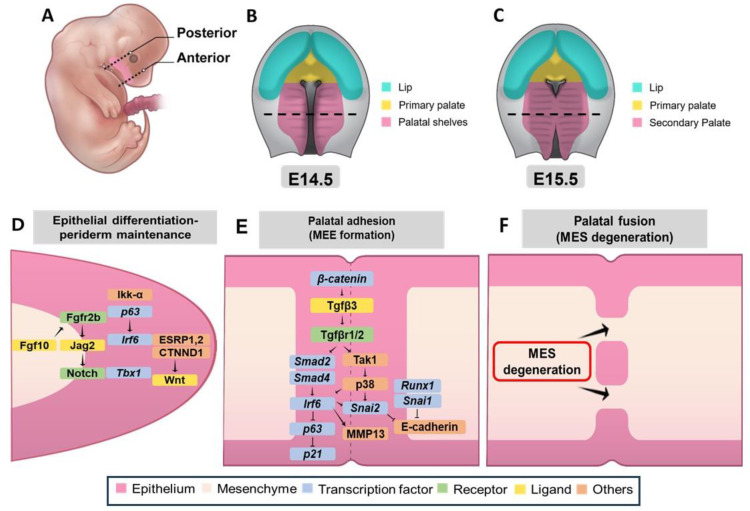
Development and molecular regulation of palatal shelf adhesion and fusion. (**A**–**C**) Schematic representation of mouse embryo development from E14.5 to E15.5. (**A**) Lateral view of E12.5 mouse embryo showing the anterior–posterior axis of palatal development. (**B**) Oral view at E14.5 showing elevated palatal shelves before fusion. (**C**) Oral view at E15.5 showing palatal fusing. (**D**–**F**) Molecular pathways controlling three key stages of palatal fusion. (**D**) Epithelial differentiation and periderm maintenance pathway showing genetic interactions. (**E**) Palatal adhesion and medial edge epithelium (MEE) formation pathway involving β-catenin, Tgf-β3, and downstream effectors. (**F**) Midline epithelial seam (MES) degeneration process leading to palatal fusion. Color code: pink—epithelium; apricot—mesenchyme; blue—transcription factors; green—receptors; yellow—ligands; orange—other regulatory molecules; black dotted line—remaining MES during palatal fusion.

**Figure 5 ijms-26-01382-f005:**
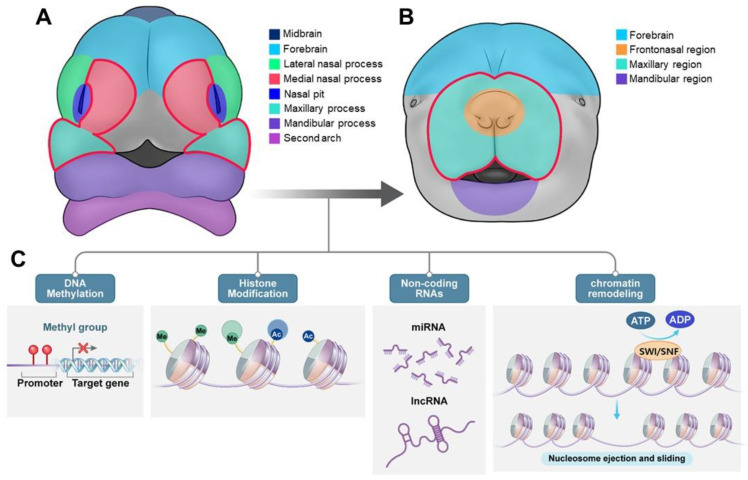
Epigenetic regulation during craniofacial development. (**A**,**B**) Schematic representation of early craniofacial development showing the morphological changes from initial facial prominences to their fusion. (**A**) The left panel shows the initial facial prominences, including midbrain, forebrain, lateral and medial nasal processes, nasal pit, maxillary and mandibular processes, and second arch. (**B**) The right panel shows the subsequent development of the frontonasal region, maxillary region, and the mandibular region. (**C**) Four major epigenetic mechanisms regulating craniofacial development: DNA methylation: addition of methyl groups to promoter regions controlling target gene expression. The red cross symbol means that the expression of the target gene is suppressed. Histone modification: post-translational modifications, including methylation (Me) and acetylation (Ac) of histone proteins. Non-coding RNAs: involvement of microRNAs (miRNA) and long non-coding RNAs (lncRNA) in gene regulation. Chromatin remodeling: ATP-dependent nucleosome ejection and sliding mediated by SWI/SNF complexes. Color code: dark blue—midbrain; light blue—forebrain; green—lateral nasal process; red—medial nasal process; navy blue—nasal pit; turquoise—maxillary process; purple—mandibular process and second arch; orange—frontonasal region; gray—other facial region behind maxillary/mandibular regions.

**Figure 6 ijms-26-01382-f006:**
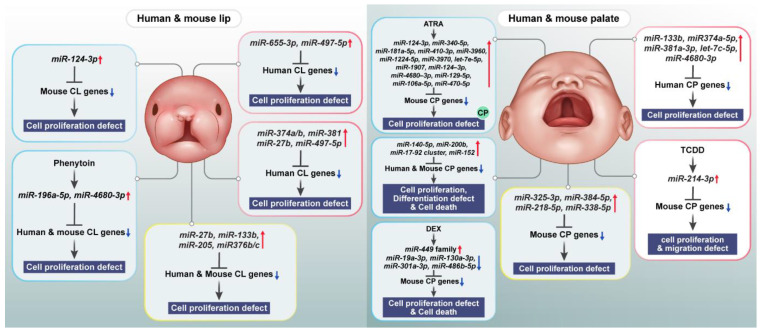
Summary of the miRNAs and genes associated with cleft lip and/or palate (CL/P) in mice and human. The complex miRNA-mediated regulatory networks involved in cleft lip and/or palate (CL/P). Environmental factors also affect cleft palate development by modulating miRNA activity, and how their dysregulation contributes to cleft formation through cell proliferation defects, differentiation defects, and cell death. Red arrows; graph showing genes going up, blue arrows; graph showing genes going down. CL, cleft lip; CP, cleft palate;

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
