# Peer review of "Molecular Regulation of Palatogenesis and Clefting: An Integrative Analysis of Genetic, Epigenetic Networks, and Environmental Interactions"

_ijms, 2025, doi:10.3390/ijms26031382_

Round 1

Reviewer 1 Report

Comments and Suggestions for Authors

The review by Hyuana et al. provides a comprehensive and detailed overview of the etiological factors underlying orofacial clefts and the molecular mechanisms involved in orofacial tissue development. It offers an engaging summary of genes critical for palatal development, particularly during secondary palate formation and fusion. Notably, the section on microRNA and epigenetics is insightful, summarizing a significant body of prior research in this rapidly evolving field of molecular biology.

However, one limitation of the study is its extensive length and the overwhelming amount of presented data. I suggest omitting the section discussing various syndromes that include facial clefts as part of their pathology. This portion detracts from the focus and does not provide substantial value to the review. Instead, the emphasis should remain on differentiating syndromic versus non-syndromic orofacial clefts.

Additionally, the authors overlooked a critical enhancer element, MCS9.7, within the IRF6 gene. This enhancer contains common and rare DNA mutations implicated in van der Woude syndrome and non-syndromic cleft lip and palate. Numerous studies have explored this regulatory element and its associated mutations, including Rahimov et al. (2008, Nature Genetics; 2024, Nature Communications), Fakhouri et al. (2012, Developmental Dynamics; 2014, HMG), Seaberg et al. (2013, CPCJ), and Kousa et al. (2018, Journal of Investigative Dermatology).

Furthermore, the genetic community now recommends using the term "22q11.2 microdeletion syndrome" instead of "DiGeorge syndrome," as regions and genes beyond TBX1 contribute to the condition.

Lastly, while the abstract and conclusion effectively discuss the interplay between genetic, epigenetic, and environmental factors in causing orofacial clefts, the study does not sufficiently explore how environmental factors influence specific genes and pathways to increase risk. This gap represents an important area for further investigation.

Author Response

reviewer1 comments

comments 1 : However, one limitation of the study is its extensive length and the overwhelming amount of presented data. I suggest omitting the section discussing various syndromes that include facial clefts as part of their pathology. This portion detracts from the focus and does not provide substantial value to the review. Instead, the emphasis should remain on differentiating syndromic versus non-syndromic orofacial clefts.

Response 1: Thank you for your valuable feedback. We understand your concern about the length of the manuscript. However, we would like to clarify that all the syndromes discussed except Tatton-Brown-Rahman syndrome (TBRS) and Arboleda-Tham syndrome include cleft palate as a major phenotype. We believe that retaining this section will enhance the completeness of the review as these syndromes are closely related to the topic. Nevertheless, we have carefully revised and summarized this section to improve clarity and focus on distinguishing between syndromic and non-syndromic orofacial clefts. Accordingly, we have removed the paragraphs on Tatton-Brown-Rahman syndrome (TBRS) and Arboleda-Tham syndrome, and have also removed the sentences related to Arboleda-Tham syndrome. We appreciate your suggestions and believe that the revisions will strengthen the manuscript.”

comments 2 : Additionally, the authors overlooked a critical enhancer element, MCS9.7, within the IRF6 gene. This enhancer contains common and rare DNA mutations implicated in van der Woude syndrome and non-syndromic cleft lip and palate. Numerous studies have explored this regulatory element and its associated mutations, including Rahimov et al. (2008, Nature Genetics; 2024, Nature Communications), Fakhouri et al. (2012, Developmental Dynamics; 2014, HMG), Seaberg et al. (2013, CPCJ), and Kousa et al. (2018, Journal of Investigative Dermatology).

Response 2: Thank you for highlighting this important point. We recognize the importance of an important enhancer element within the IRF6 gene, MCS9.7, in relation to van der Wood syndrome and non-syndromic cleft lip and palate. To address this, we have included the following information on mutations associated with MCS9.7 “Given the critical role of Irf6 in these processes, the MCS-9.7 regulatory region may act as a mutational hotspot for both rare and common genetic variations. These variations could lead to or increase the risk of different forms of orofacial clefts (OFCs) by disrupting Irf6 expression in the periderm or basal layers of the oral epithelium.” We appreciate your insightful suggestions, which have improved the completeness of our review.

comments 3 : Furthermore, the genetic community now recommends using the term "22q11.2 microdeletion syndrome" instead of "DiGeorge syndrome," as regions and genes beyond TBX1 contribute to the condition.

Response 3 : Thank you for your suggestion. We acknowledge the updated terminology recommendation within the genetic community. Accordingly, we have replaced "DiGeorge syndrome" with "22q11.2 microdeletion syndrome" throughout the manuscript to align with current standards. We appreciate your feedback, which has helped improve the accuracy and clarity of our work.

comments 4 : Lastly, while the abstract and conclusion effectively discuss the interplay between genetic, epigenetic, and environmental factors in causing orofacial clefts, the study does not sufficiently explore how environmental factors influence specific genes and pathways to increase risk. This gap represents an important area for further investigation.

Response 4 : Thank you for your valuable comments on environmental factors. We have supplemented the content and references as per the reviewer’s suggestions. What has changed is as follows:

  • ATRA:

“{Wei, 2018} has showed that ATRA administration induced changes in miRNA in mice with CP. The CP (CP) in RA-treated mice has been linked to the upregulation of miR-181a-5p, miR-410-3p, miR-3960, miR-1224-5p, miR-3970, let-7e-5p, miR-1907, miR-124–3p and miR-4680–3p {Wei, 2018} [184]. Then, {Wei, 2018} demonstrated that miR-140-3p, miR-351-5p, and miR-503-5p were downregulated in a mouse model of RA-treated (Figure 6).”, “Also, the previous study showed that miR-129-5p and miR-340-5p repress cell growth of primary mouse embryonic palatal mesenchymal cells and neural crest-derived O9-1 cells. Specifically, miR-129-5p targets Sox5 and Trp53, while miR-340-5p regulates the expression of Chd7, Fign, and Tgfbr1 to modulate cell growth {Yoshioka, 2022}.”

  • TCDD

“Recently, the upregulation of miRNAs (miR-214-3p, miR-296-5p, and miR-33-5p) in the TCDD-treated group, while miRNAs(miR-92a-3p, miR-126a-3p, and miR-411-5p) were sig-nificantly downregulated. Notably, qRT-PCR testing confirmed a significant difference in miR-214-3P expression. These findings support that TCDD inhibits palatal mesenchymal cell proliferation and migration through miR-214-3p, suggesting a its role in TCDD-induced CP in mice {Dong, 2024 #621}.”

  • Folate

“In study, {Gonseth, 2019} tested the hypothesis that the prevention by folic acid is an epigenetic modification, specifically by determining whether changes in DNA methylation are associated with CL/P. Using the Illumina® Human Beadchip 450K array, Genomic DNA methylation levels were assessed from archived newborn bloodspots in a 182-member case-control study. CL/P cases demonstrated significant epigenome-wide hypomethylation compared to controls; 63% of CpG sites showing reduced methylation in case new-borns. his was consistent across racially stratified subgroups. A total of 28 CpG sites reached epigenome-wide significance, all showing hypomethylation in cases. The most significant CL/P-associated differentially methylated region included the VTRNA2-1 gene, which was also hypomethylated in cases (FWER p = 0.014) {Gonseth, 2019}. This region has been previously described as a nutritionally-responsive, metastable epiallele. CL/P-associated methylation changes were generally more pronounced at or near putative metastable epi-allelic regions. Gene Set Enrichment Analysis of CL/P-associated DMRs revealed an over-representation of genes involved in palate development, such as WNT9B, MIR140, and LHX8. These DNA methylation changes may partly explain the mechanism by which OFCs are responsive to maternal folate levels {Gonseth, 2019}.”

  • Phenytoin

“Phenytoin at teratogenic doses reduces the rate of the early embryonic heart of pregnant rats and causes an increase in the length of time embryonic hypoxia is sustained, which would explain the observed findings that hypoxia elevates and hyperoxia lowers the incidence of CL in sensitive mouse strains. The mechanism underlying this effect has been suggested to be the phenytoin's inhibition of the HERG potassium channel along with its reduction of sodium and calcium channels which could also explain the action on the embryonic heart {William, 2006}.”, “In addition, {William, 2006} found that phenytoin, an inducer of CL, decreased cell proliferation through miR-196a-5p induction. During E10.5 to E12.5, miR-196a-5p expression significantly declines in the MxPs and NPs, while phenytoin upregulates miR-196a-5p and sup-presses MELM cell proliferation by downregulating Pbx1, Pbx3, and Rpgrip1l {Iwaya, 2023}. Especially, treatment with a specific inhibitor for miR-196a-5p restored cell proliferation through normalization of expression of CL-associated genes in the cells treated with phenytoin {William, 2006}. miR-196a-5p suppresses cell proliferation and promotes osteogenic differentiation in human Wharton’s jelly umbilical cord stem cells (WJCMSC) and sup-presses bone formation in WJCMSC-sheet transplanted rat calvaria through suppression of Serpinb2 {Iwaya, 2023}. These collective findings suggest a pleiotropic role for miR-196a-5p in diverse developmental processes, including those critical for palate formation. Recently, {Tsukiboshi, 2024} have shown that phenytoin inhibits HEPM cell proliferation in a dose-dependent manner by downregulating cyclin-D1 and cyclin-E. phenytoin treatment upregulated miR-4680-3p while downregulating its target genes, ERBB2 and JADE1. These findings indicate that phenytoin suppresses cell proliferation by modulating miR-4680-3p {Tsukiboshi, 2024}.”

Reviewer 2 Report

Comments and Suggestions for Authors

The authors reviewed lip and palate formation and the reason to induce CL/P showing molecular mechanism, epigenetic, and environmental factors.

I feel this review helps to understand craniofacial research for beginner. Therefore, I would like to update below comment to accept.

Major

1)     In figure 6, the reader will be confusing. Although the left figure showed mouse lip, the result is mixing mouse and human CL. In the right figure is same. The authors should modify the figure. In addition, DEX is reported to decrease miR-130a-3p. Moreover, the authors need to add phenytoin in the section of atRA (miR-4680-3p ) as well.

2)     The authors seemed to miss [PMID 35774010]. I think the authors should mention this manuscript to review miRNA.

3)     atRA-induced miRNA expression report seemed to be missing. The authors should add entitled “Identification of the Differentially Expressed microRNAs Involved in Cleft Palate Induced by Retinoic Acid (RA) in Mouse Model” around line 1292.

4)     Around 1374, recent investigation revealed that TCDD suppressed cell proliferation and migration through miR-214-3p [PMID 39314083]. The authors should add latest information.

5)     Around 1473, recent investigation revealed that phenytoin suppressed cell proliferation and migration through miR-4680-3p [PMID 38191190]. The authors should add latest information.

6)     Table 3. Let-7c-5p, miR-193a-3p. If the authors would like to add “hsa”, the other miRNA should add species. In addition, gene information is missing in the present table. Moreover, reference seemed to be not correct. The authors should check carefully.

7)     Line 1034, not only ref 184 but also adding 180 is correct.

8)     Line 1039, I could not understand how seven miRNAs were identified. The authors need to show seven miRNAs correctly. In lilne 1040, “we” seemed to be not correct. The authors should change [PMID 38241808].

Minor

1)     I feel line 86 “E17” seemed to be E15.5.

2)     Line 95, ref 138 is correct ? If correct, the authors should change reference numbering.

3)     CP seemed to change cleft palate throughout the manuscript. The authors should change line 219, 996, 1035, 1135, 1279, 1291, 1402.

4)     In line 1040, what PB means ?

5)     In line 1280, miR-4680-3p seemed to be not correct. The authors need to change miR-340-5p and cite [PMID 35420127]

6)     The authors mixed E and GD. I recommend to unify E (line 1021, 1060, 1068, 1102).

7)     The line 1469 and 1471, the authors are showing miss information. The authors should cite [PMID 335721377].

8)     Abbreviation is so missing. The authors should correct as below.

CL/P : line 79, 444, 445. 915, 1424, 1494, 1511

FGF: line 312, 1356

TGF: line 411, 1354, 1488, 1489

BMP: line 413, 1489

IRF6: line 656

VWS: line230, 233, 657

WHS: line 281

miRNA: line 688, 980

OFC: line 700, 775, 820, 822, 873, 1228, 1423, 1487

CLP: line 700. 849

CPO: line 700, 850, 1303, 1477, 1495

EWAS: line 820, 821

CLO: line 850

NCC: line 876, 885, 902, 945, 958, 960, 993, 1008, 1169, 1192, 1219

atRA: 955, 1260, 1296

EMT: line 989, 1087, 1159,

SATB2: line 1207

RA: line 1306

AcAL: line 1309

TCDD: line 1332

MEE: line 1342

AhR: line 1353

Author Response

Reviewer2 comments

Major

Comments 1 :  In figure 6, the reader will be confusing. Although the left figure showed mouse lip, the result is mixing mouse and human CL. In the right figure is same. The authors should modify the figure. In addition, DEX is reported to decrease miR-130a-3p. Moreover, the authors need to add phenytoin in the section of atRA (miR-4680-3p ) as well.

Response 1 : Thank you for the detailed comments from the reviewer. In Figure 6, we added some miRNAs to the TCDD and DEX parts as follows, and also modified the title part of the figure. The parts added according to the reviewer's comments were also included in the figure. Thanks to your comments, we were able to add other parts as well.

Comments 2 :  The authors seemed to miss [PMID 35774010]. I think the authors should mention this manuscript to review miRNA.

Response 2 : Thank you for your insightful comments. We acknowledge that our study did not include the review miRNA. So we added the content based on your advice. “{Yan, 2022} demonstrated that CP model gene and miRNA expression from E10.5 to E14.5 in the maxillary processes to identify spatiotemporal patterns of gene and miRNA expression. MiR-325-3p and miR-384-5p, that repressed cleft-related genes Adamts3, Runx2, Fgfr2, Acvr1, and Edn2, while their expression increased over time. On the contrary, miR-218-5p and miR-338-5p, repressed cleft-related genes Pbx2, Ermp1, Snai1, Tbx2, and Bmi1, while their expression decreased over time. The findings suggest that these miRNA mimics sig-nificantly inhibited the cell proliferation of the mouse palate mesenchymal cell and O9-1 cranial neural crest cell line by modulating CP-associated genes, confirming their regula-tory role in the pathogenesis {Yan, 2022}.”

Comments 3 : atRA-induced miRNA expression report seemed to be missing. The authors should add entitled “Identification of the Differentially Expressed microRNAs Involved in Cleft Palate Induced by Retinoic Acid (RA) in Mouse Model” around line 1292.

Response 3 : We appreciate your attention to detail, which has helped improve the clarity and accuracy of this manuscript. The corrections are as follows: “{Wei, 2018} has showed that ATRA administration induced changes in miRNA in mice with CP. The CP (CP) in RA-treated mice has been linked to the upregulation of miR-181a-5p, miR-410-3p, miR-3960, miR-1224-5p, miR-3970, let-7e-5p, miR-1907, miR-124–3p and miR-4680–3p {Wei, 2018} [184]. Then, {Wei, 2018} demonstrated that miR-140-3p, miR-351-5p, and miR-503-5p were downregulated in a mouse model of RA-treated (Figure 6).”

Comments 4 : Around 1374, recent investigation revealed that TCDD suppressed cell proliferation and migration through miR-214-3p [PMID 39314083]. The authors should add latest information.

Response 4 : Thank you for your insightful advice. As reviewer’s suggestion, we have included references as follows. “Recently, the upregulation of miRNAs(miR-214-3p, miR-296-5p, and miR-33-5p) in the TCDD-treated group, while miRNAs(miR-92a-3p, miR-126a-3p, and miR-411-5p) were significantly downregulated. Notably, qRT-PCR testing confirmed a significant difference in miR-214-3P expression. These findings support that TCDD inhibits palatal mesenchymal cell proliferation and migration through miR-214-3p, suggesting a its role in TCDD-induced cleft palate in mice {Dong, 2024}.”

Comments 5 : Around 1473, recent investigation revealed that phenytoin suppressed cell proliferation and migration through miR-4680-3p [PMID 38191190]. The authors should add latest information.

Response 5 : We have revised the statement to more accurately reflect our reviews. The revised text reads: “Recently, {Tsukiboshi, 2024} have shown that phenytoin inhibits HEPM cell proliferation in a dose-dependent manner by downregulating cyclin-D1 and cyclin-E. phenytoin treatment upregulated miR-4680-3p while downregulating its target genes, ERBB2 and JADE1. These findings indicate that phenytoin suppresses cell proliferation by modulating miR-4680-3p {Tsukiboshi, 2024}.”

Comments 6 : Table 3. Let-7c-5p, miR-193a-3p. If the authors would like to add “hsa”, the other miRNA should add species. In addition, gene information is missing in the present table. Moreover, reference seemed to be not correct. The authors should check carefully.

Response 6 : Thank you for your insightful advice. We have corrected the part about miRNA of the table (“let-7c-5p, miR-193a-3p”) and reference. Thank you for your comments helped me find additional part to supplement.

Comments 7 : Line 1034, not only ref 184 but also adding 180 is correct.

Response 7 : Thank you for your accurate advice. We have revised the statement to more accurately reflect our reviews. “In addition, {Suzuki, 2019} and [186] identified miRNAs (miR-133b, miR-140-5p, miR-374a-5p, miR-381a-3p, and miR-4680-3p) associated with CP development in humans through sys-tematic reviews, bioinformatics analyses, and cell proliferation assays in human embry-onic palatal mesenchymal(HEPM) cells[182, 186].”

Comments 8 : Line 1039, I could not understand how seven miRNAs were identified. The authors need to show seven miRNAs correctly. In lilne 1040, “we” seemed to be not correct. The authors should change [PMID 38241808].

Response 8 : We totally understood the reviewer’s comment. We have corrected the sentences as follows. “[185] has showed that Genes predicted in all three databases (FUNRICH, MIRDB, and Targetscan) were considered as potential target genes of the miRNAs (104 target genes of miR-193a-5p, 512 target genes of let-7c-5p). In particular, PIGA and TGFB2 were selected as the most promising targets by further inquiring into three databases (MGI, MalaCards, and DECIPHER). Recently, {Tsukiboshi, 2024} found that Phenobarbital(PB) specifically induced let-7c-5p expression and that the let-7c-5p specific inhibitor alleviated the PB-induced suppression of HEPM cell proliferation, indicating that let-7c-5p plays a crucial role in PB-induced tox-icity. Let-7c-5p is highly expressed in craniofacial tissues of embryonic mice {Tsukiboshi, 2024}.”

Minor

Comments 1 : I feel line 86 “E17” seemed to be E15.5.

Response 1 : Thank you for pointing this out. We agree with your comment. So to clarify, I've revised the text to read “In mice, this process starts at approximately E11.5 and is essentially completed by E15.5.”

Comments 2 : Line 95, ref 138 is correct ? If correct, the authors should change reference numbering.

Response 2 : Thank you for your careful review. We have verified that the reference is correct. Therefore, we have adjusted the reference number throughout the manuscript to ensure consistency and accuracy. We appreciate your attention to detail that has helped improve the clarity of the work.

Comments 3 : CP seemed to change cleft palate throughout the manuscript. The authors should change line 219, 996, 1035, 1135, 1279, 1291, 1402.

Response 3 : Thank you for your valuable comments. We have carefully reviewed the manuscript and revised the first occurrence of the term to the full term and then revised subsequent occurrences to the abbreviated term to ensure clarity and consistency. Additionally, based on your insightful comment, we have also corrected similar terminology in other lines that were not specifically mentioned. All modifications have been marked in red in the revised manuscript for easy identification. We truly appreciate your keen insight and thoughtful suggestions. This has greatly helped improve the clarity and precision of our manuscript.

Comments 4 : In line 1040, what PB means ?

Response 4 : We appreciate your attention to detail, which has helped improve the clarity and accuracy of this manuscript. As reviewer’s suggestion, we have added the sentences as follows. Recently, {Tsukiboshi, 2024} found that Phenobarbital(PB) specifically induced let-7c-5p expression and that the let-7c-5p specific inhibitor alleviated the PB-induced suppression of HEPM cell proliferation, indicating that let-7c-5p plays a crucial role in PB-induced tox-icity. Let-7c-5p is highly expressed in craniofacial tissues of embryonic mice {Tsukiboshi, 2024}.”

Comments 5 : In line 1280, miR-4680-3p seemed to be not correct. The authors need to change miR-340-5p and cite [PMID 35420127]

Response 5 : Thank you for your insightful advice. As per the reviewer's suggestion, I have reordered the references to fit the content. “Also, the previous study showed that miR-129-5p and miR-340-5p repress cell growth of primary mouse embryonic palatal mesenchymal cells and neural crest-derived O9-1 cells. Specifically, miR-129-5p targets Sox5 and Trp53, while miR-340-5p regulates the expression of Chd7, Fign, and Tgfbr1 to modulate cell growth {Yoshioka, 2022}.”

Comments 6 : The authors mixed E and GD. I recommend to unify E (line 1021, 1060, 1068, 1102).

Response 6 : We appreciate reviewer’s detailed comments. We have corrected the part about embryonic day in text. “In particular, strong expression was found in the maxillary process of miR-124-3p, particularly at E13.5.”, “A decrease in the expression of miR-17-92 from E12-14, with a concomitant increase in the expression of key components of the pathway, such as TGFBR2, SMAD2, and SMAD4, was evident in the palatal shelves.”, “The findings in palatal mesenchymal cells (PMCs) include the expression of E2F1 and E2F3 in palatal tissue on E12-14, significant inhibition of cell proliferation upon E2F1 knockdown, and upregulation of the miR-17-92 cluster with E2F1 overexpression [190].”, “Analysis of the MEE of Tgf-β3-/- mouse fetuses at E13.5, indicated that the expression of miR-206 in the developing palate was significantly diminished compared to that in their wild-type counterparts, suggesting an important role of miR-206 in palatal ontogeny [15] [195].”

Comments 7 : The line 1469 and 1471, the authors are showing miss information. The authors should cite [PMID 335721377].

Response 7 : We totally agree with reviewer’s comment. We changed the part as below. In addition, following the reviewer's advice, we have supplemented any parts that require further explanation. “In addition, {William, 2006} found that phenytoin, an inducer of CL, decreased cell proliferation through miR-196a-5p induction. During E10.5 to E12.5, miR-196a-5p expression significantly declines in the MxPs and NPs, while phenytoin upregulates miR-196a-5p and sup-presses MELM cell proliferation by downregulating Pbx1, Pbx3, and Rpgrip1l {Iwaya, 2023}. Especially, treatment with a specific inhibitor for miR-196a-5p restored cell proliferation through normalization of expression of CL-associated genes in the cells treated with phenytoin {William, 2006}. miR-196a-5p suppresses cell proliferation and promotes osteogenic differentiation in human Wharton’s jelly umbilical cord stem cells (WJCMSC) and sup-presses bone formation in WJCMSC-sheet transplanted rat calvaria through suppression of Serpinb2 {Iwaya, 2023}. These collective findings suggest a pleiotropic role for miR-196a-5p in diverse developmental processes, including those critical for palate formation. Recently, {Tsukiboshi, 2024} have shown that phenytoin inhibits HEPM cell proliferation in a dose-dependent manner by downregulating cyclin-D1 and cyclinE. phenytoin treatment upregulated miR-4680-3p while downregulating its target genes, ERBB2 and JADE1. These findings in-dicate that phenytoin suppresses cell proliferation by modulating miR-4680-3p {Tsukiboshi, 2024}.”

Comments 8 : Abbreviation is so missing. The authors should correct as below.

  • CL/P: line 79, 444, 445. 915, 1424, 1494, 1511
  • FGF : line 312, 1356
  • TGF : line 411, 1354, 1488, 1489
  • BMP : line 413, 1489
  • IRF6: line 656
  • VWS : line230, 233, 657
  • WHS: line 281
  • miRNA: line 688, 980
  • OFC : line 700, 775, 820, 822, 873, 1228, 1423, 1487
  • CLP : line 700. 849
  • CPO : line 700, 850, 1303, 1477, 1495
  • EWAS: line 820, 821
  • CLO: line 850
  • NCC: line 876, 885, 902, 945, 958, 960, 993, 1008, 1169, 1192, 1219
  • atRA: 955, 1260, 1296
  • EMT: line 989, 1087, 1159,
  • SATB2: line 1207
  • RA: line 1306
  • AcAL: line 1309
  • TCDD: line 1332
  • MEE : line 1342
  • AhR: line 1353

Response 8: Thank you for your valuable comments. We have carefully reviewed the manuscript and revised the first occurrence of each term to its full term, and then revised subsequent occurrences to the abbreviated term to ensure clarity and consistency. We have also revised similar terms in other lines that were not specifically mentioned based on your insightful comments. All revisions have been highlighted in red in the revised manuscript for easy identification. We sincerely appreciate your keen insight and thoughtful suggestions, which have greatly helped us improve the clarity and accuracy of the manuscript.

Round 2

Reviewer 1 Report

Comments and Suggestions for Authors

The authors did a great job of improving the quality of the review manuscript. I have no further comments or suggestions.